

# Identifying 'persistent temperature inversion' events in a Subalpine Basin using Radon-222

Dafina Kikaj[1], Janja Vaupotič[2], and Scott D. Chambers[3]

[1]Jožef Stefan International Postgraduate School, Jamova cesta 39, 1000 Ljubljana, Slovenia
[2]Jožef Stefan Institute, Department of Environmental Sciences, Jamova cesta 39, 1000 Ljubljana, Slovenia
[3]ANSTO, Environmental Research, Locked Bag 2001, Kirrawee DC, NSW 2232, Australia

*Correspondence to*: Dafina Kikaj (dafina.kikaj@ijs.si)

**Abstract.** One year of meteorological and atmospheric radon observations in a topographically-complex Subalpine Basin are used to identify 'persistent temperature inversion' (PTI) events. PTI events play a key role in public health due to the accumulation of urban pollutants that they cause. Two identification techniques are compared: a new method, based on single-height radon measurements from a single centrally-located station, and an existing approach based on observations from eight weather stations around the Subalpine Basin. After describing the radon-based method (RBM), its efficacy is compared with that of the existing pseudo-vertical temperature gradient method (TGM). The RBM identified 6 PTI events over the year (4 in winter, 2 in autumn), a subset of the 17 events identified by the TGM. The RBM is demonstrated to be more consistent in its identification of PTI events, and more selective of persistent strongly stable conditions. Furthermore, its performance is seasonally independent. The comparatively poor performance of the TGM was attributed to seasonal inconsistencies in the validity of the method's key assumptions (influenced by mesoscale processes, such as local drainage flows, nocturnal jets, and intermittent turbulence influence), and a lack of snow cover in the basin for the 2016−2017 winter period. Corresponding meteorological quantities for RBM PTI events (constituting 27 % of the autumn-winter "cold season"), were well characterised. PTI wind speeds in the basin were consistently low over the whole diurnal cycle (typically 0.2–0.6 m s$^{-1}$). The comparative efficacy of the RBM for PTI air quality assessment is demonstrated using hourly $PM_{10}$ observations throughout the year. Peak hourly mean $PM_{10}$ concentrations for winter (autumn) PTI events were underestimated by 13 μg m$^{-3}$ (11 μg m$^{-3}$) by the TGM compared with the RBM. Only the RBM indicated that nocturnal hourly mean $PM_{10}$ values in winter PTI events can exceed 100 μg m$^{-3}$, the upper threshold of low-level short-term $PM_{10}$ exposure according to World Health Organisation guidelines. The efficacy, simplicity and cost effectiveness of the RBM for identifying PTI events has the potential to make it a powerful tool for urban air quality management in complex terrain regions; for which it adds an additional dimension to contemporary atmospheric stability classification tools. Furthermore, the long-term consistency of the radon source function will enable the RBM to be used in the same way in future studies, enabling the relative magnitude of PTI events to be gauged, which is expected to assist with the assessment of public health risks.

## 1 Introduction

Urban air pollution is one of the main environmental health risks in Europe (EEA, 2017; WHO, 2014). It derives from aerosols and/or trace gases of natural (e.g., sandstorms, volcanos, forest fires, and ocean spray) or



anthropogenic origin (e.g., biomass burning, power generation, transport, and industry) (Beeston et al., 2011; Ferrario et al., 2008; Langmann et al., 2012; Miranda et al., 2015; Pey et al., 2013; Vardoulakis and Kassomenos, 2008). Public health effects can be acute or chronic in nature, and include: respiratory illnesses, cardiovascular disorders, conjunctivitis, skin irritations, meningococcal meningitis, valley fever, and even
mortality (Goudie, 2014; Griffin, 2005; Highwood and Kinnersley, 2006; Pope and Dockery, 2006). With public health in mind, the World Health Organization (WHO) identified numerous 'criteria pollutants' in urban regions for which they have recommended guidelines on concentrations and/or duration (the Air Quality Directive; EEA, 2017). An 'air pollution event' is considered to have occurred when the concentration of one or more of the criteria pollutants exceeds the guideline value for longer than the specified duration threshold. Such
pollution events can be of short duration (1−12 hours, day or night) or, under certain conditions, they can persist for days to weeks.

Urban air quality can be influenced by remote, regional, or local processes, depending on the prevailing meteorology (Kukkonen et al., 2005). The particular focus of this study is on pollution events arising as a result of local anthropogenic emissions, the effects of which are exacerbated under specific synoptic weather
conditions: cold-season anticyclones. Anticyclonic weather conditions are associated with relatively calm winds and clear skies, due to regional scale subsidence. In summer, anticyclonic conditions can result in daytime pollution events as a result of rapid secondary-pollutant formation, or night time pollution events, when strong radiative cooling results in the formation of shallow, poorly mixed 'inversion layers'. Such pollution events usually end abruptly each morning shortly after the onset of convective mixing as a result of intense sunlight on
dry surfaces. In such cases, near-surface concentrations of primary emissions drop ~50% from peak values by mid-morning and reach their minimum values in the early afternoon when the atmospheric boundary layer (ABL) is deep and uniformly mixed (Chambers et al., 2015). In winter, however, day lengths are shorter, sun angles lower, and surfaces can be wet or frozen. In topographically complex regions, when wind speeds and day length can be even further reduced, and katabatic flows down the basin walls result in the 'pooling' of cold air,
largescale subsidence associated with slow-moving or stagnant anticyclonic systems can inhibit the daytime mixing that usually disperses accumulated nocturnal pollution, resulting in air pollution events that can persist for days to weeks (Cuxart et al., 2006; Kassomenos and Koletsis, 2005). Here we refer to such extended periods of severe air pollution as 'persistent temperature inversion' (PTI) events.

In light of the above, the ability to reliably identify PTI events and objectively quantify their magnitude is
30 important for improved understanding of pollutant concentration variability, assessing potential health impacts on residents, and developing new mitigation measures for air pollution in such complex topographical regions. Unfortunately, there have been relatively few observational studies of PTI events (Clements et al., 2003; Dorninger et al., 2011; Lareau et al., 2013; Lehner et al., 2015; Silcox et al., 2012). To date, this lack of observational studies has mainly been due to the difficulty in making the necessary meteorological
measurements within the topographically-challenging regions (Yao and Zhong, 2009). Furthermore, such studies often require the deployment of sophisticated instrumentation (e.g., radiosondes, tethered balloons, masts or aircraft), which are typically expensive and labour intensive to operate.

The first European field campaign to target PTI events took place in the Chamonix-Mont-Blanc valley, in January–February 2015. This study was motivated by the anomalously high pollution concentrations frequently
recorded there. The comprehensive suite of meteorological and chemical observations made throughout the



campaign has been described by Chemel et al. (2016). A meteorological approach, based on differences between air temperatures from automatic weather stations at various elevations of the sidewalls of the valley, was used to characterize the stable nocturnal conditions (Dorninger et al., 2011; Whiteman et al., 2004) based on the assumption of horizontally homogenous air temperature. The same pseudo-vertical temperature gradient method

was employed by Largeron and Staquet, (2016a) in the French Alps to detect PTI events throughout the cold season (November 2006 to February 2007). There are, however, a variety of mesoscale processes that have the potential to disrupt the vertical temperature profile and its horizontal homogeneity (including local drainage flows, nocturnal jets, and intermittent turbulence) (Whiteman et al., 2004; Williams et al., 2013). Such influences can complicate the interpretation of pseudo-vertical temperature gradients derived from stations of

contrasting elevations on valley or basin walls, depending on their particular spatial locations.

By contrast, measurement of a surface-emitted atmospheric tracer with appropriate physical properties (e.g., simple source and sink characteristics), which responds directly to atmospheric mixing processes, has the potential to provide a more consistent and representative method by which to identify PTI events. Such a method should be uniformly applicable, allowing seasonal changes in the number and duration of such events to

be determined. Furthermore, once a representative number of events have been identified, seasonally-dependent threshold concentrations of the tracer could be determined to help gauge the severity of the inversion events and characterise the meteorological conditions with which they are associated. The naturally-occurring radioactive gas radon ($^{222}$Rn) is an ideal candidate for this task. The use of radon as a tracer in atmospheric studies dates back from the early 1900s (Eve, 1908; Satterly, 1910; Wigand and Wenk, 1928; Wright and Smith, 1915). In

particular, however, radon has achieved considerable credibility in the field of atmospheric science as an indicator of vertical mixing and transport near the Earth's surface from 1960s until nowadays (Moses et al., 1960; Kirichenko, 1962; Cohen et al., 1972; Allegrini et al., 1994; Perrino et al., 2001; Sesana et al., 2003; Galmarini, 2006; Avino and Manigrasso, 2008; Chambers et al., 2011; Williams et al., 2011, 2013; Pitari et al., 2014; Chambers et al., 2015; Wang et al., 2016; Williams et al., 2016; Chambers et al., 2018).

Radon concentrations exhibit a high degree of variability on hourly to seasonal timescales. The majority of this variability is attributable to processes occurring at diurnal, synoptic and seasonal timescales. Other influences, such as those arising from mesoscale motions, or the effects of geographical variability in soil characteristics, rainfall, wind speed and atmospheric pressure on the radon source function (Chambers et al., 2011; Etiope and Martinelli, 2002; Karstens et al., 2015; Levin et al., 2002; Mazur and Kozak, 2014), are of secondary

significance and random in nature, so they typically average out in long-term (seasonal) statistics. The PTI identification technique developed in this study necessitates the ability to characterise and separate the different temporal scales contributing to the variability observed in a radon time series.

**Seasonal variability** of radon concentrations is contributed to by long-term changes in (i) air mass fetch, (ii) the radon source function (through average soil moisture, snow cover or soil freezing), (iii) day length, and (iv) the

mean atmospheric boundary layer depth (higher in summer, lower in winter), brought about by the annual solar cycle. **Synoptic variability** (timescale ≤ 2 weeks) is mainly attributable to short-term changes in air mass fetch, boundary layer depth and ventilation of the boundary layer, brought about by the passage of synoptic weather systems. In this case, variability of the radon source function over the air mass' recent fetch can have a greater influence on observed concentrations than the local radon source function at the measurement site. **Diurnal**



**variability**, on the other hand, is primarily attributable to changes in the atmospheric mixing state (i.e., mixing depth or "stability"), and the strength of the *local* radon source function.

The diurnal radon cycle is characterised by a mid-afternoon minimum (the magnitude of which is related to the air mass' fetch history) and a nocturnal maximum near dawn (when the atmospheric mixing depth is shallowest). The *amplitude* of the diurnal radon cycle is directly related to the degree to which near-surface turbulent mixing is suppressed each night. Synoptic cycles of the *afternoon minimum* radon concentrations can be analogous to the *diurnal* radon cycle, but on longer timescales. These cycles are characterised by minimum values when gradient winds are strong (or terrestrial fetch is limited), and maximum values when gradient winds are low and there is subsidence (or during extended periods of terrestrial fetch, including stagnation episodes). In the warmer months, when days are convective and nights are stable, diurnal cycle amplitudes exceed synoptic cycle amplitudes. In the colder months, however, the amplitude of the synoptic cycle can be comparable to, or even greater than, that of the diurnal cycle. This seasonal change in relative amplitudes of the diurnal and synoptic radon cycles forms the basis of our method to identify PTI events.

The aims of this study are to use the first year of continuous joint atmospheric radon and meteorological measurements in a Subalpine Basin (Ljubljana, Slovenia) to: (i) develop a seasonally-independent radon-based method to reliably identify 'persistent temperature inversion' events in complex topographic regions; (ii) compare the performance of the radon-based scheme to that of a previously published pseudo-vertical temperature gradient method for identifying PTI events over the whole year; (iii) investigate the seasonality of PTI event occurrence in the Subalpine Basin; and (iv) characterize the local meteorological conditions (air temperature, relative humidity, cloud cover, wind speed, wind direction and precipitation) associated with PTI events, as well as discuss the implications of these conditions for urban air quality using hourly urban $PM_{10}$ observations as an example.

## 2 Methodology

### 2.1 Radon as a tracer of atmospheric transport and mixing

Radon is the gaseous decay product of Radium ($^{226}$Ra), a member of the Uranium ($^{238}$U) chain, which is ubiquitous all over the Earth's crust. Radon is fairly uniformly distributed on local to regional scales ($\pm 25\%$; Jacob et al., 1997; Karstens et al., 2015), and has a source function that is 2−3 orders of magnitude greater from unsaturated land surfaces than from the open ocean (Schery and Huang, 2004). Once emitted to the atmosphere radon is directly influenced by the meteorological processes that regulate turbulent mixing, namely, vertical and horizontal components of dispersion. Being an inert gas, radon does not chemically react with other atmospheric constituents and its low solubility makes it unlikely to be washed out by rainfall. Consequently, it is eliminated from the atmosphere predominantly by radioactive decay (half-life 3.82 days). While radon's short half-life prevents it from accumulating in the atmosphere greater than synoptic timescales, it is possible to track air masses that have been in contact with terrestrial surfaces over the ocean, or within the troposphere, for 2−3 weeks, enabling it to carry important geophysical information over long distances. Since radon's half-life is much greater than mixing timescales of the atmospheric boundary layer, and since radon is still relatively conservative (within 5−10%) over the course of a whole night, radon is an appropriate atmospheric tracer for near-surface mixing studies from hourly to nightly timescales. Furthermore, based on the relatively small



fraction of terrestrial surfaces occupied by urbanisation, at the regional scale radon's terrestrial source function is not significantly affected by human activity.

### 2.2 Measurement sites and surrounds

The Ljubljana Subalpine Basin site (SAB; 46.05°N, 14.51°E; 299 m above sea level, a.s.l.) is in the capital city

of Slovenia, which has a population of around 280 000. The city's climate, classified as temperate and continental, is typical of southern European-subalpine regions. Air temperature's exhibit a distinct seasonal cycle, with daily means reaching a maximum of 22 °C in summer and a minimum of 0 °C in winter. Prevailing winds for the region are NE, NNE and ENE, and the rainfall distribution is irregular, peaking in summer (ARSO, 2017).

The SAB is surrounded by peaks of the Julian and Kamnik-Savinje (Vrhovec, 1990; Figure 1b). To the north the region is bounded by the Kamnik-Savinje Alps (*highest peaks*: Grintovec 2,558 m and Krvavec 1,742 m), to the west the Šmarna Gora and Rašica hills connect to the Polhov Gradec hills (*highest peak:* Pasja Ravan, 1,019 m), and to the east the Posavje hills (*highest peak:* Kum, 1,211 m). Immediately to the south is "Ljubljansko barje" (Šajn et al., 2011), a drained ancient moor that was frequently a lake during the ice ages (Pak, 1992).

The air quality of Ljubljana is affected by a combination of local and remote influences, the relative contributions from which are dictated by prevailing meteorology. The geographic location of Slovenia (Figure 1a) is such that remote influences on air quality, when they occur, are mainly a result of pollution transport from northern Africa (Kallos et al., 2006), the northern Balkan Peninsula, or from Eastern Europe (Poberžnik and Štrumbelj, 2016).

Drag associated with the complex topography of the surrounding region retards synoptic-scale winds, resulting in a mean near-surface annual wind speed of only 1.3 m s$^{-1}$. Extended periods of anticyclonic conditions (low-gradient winds and subsidence) are also common for the region. Combined with the unique concave shape of the subalpine Ljubljana basin (Rakovec et al., 2002), this results in comparatively calm conditions in the colder months, with frequent foggy days and 'persistent temperature inversions' (120 d y$^{-1}$), favouring the build-up of

locally-sourced emissions.

### 2.3 Measurement methods

Atmospheric radon measurements were made for 1 year (December 2016 to November 2017) at the Ljubljana Bežigrad (LjB) automatic weather station in the central part of the city (46°07' N, 14°52' E; Figure 1c), and are ongoing. The measurements were conducted using an AlphaGUARD (Bertin Instruments) ionisation chamber

operating in diffusion mode with a 1-hour integration time. The instrument was mounted 1.5 m a.g.l., within a Stevenson Screen instrument shelter. The AlphaGUARD has a lower limit of detection (LLD) of around 2 Bq m$^{-3}$. Over the year only 0.01% of observed radon concentrations were at or below this limit, which justifies the continued use of the AlphaGUARD at this site as a tool for air quality assessment.

The LjB weather station, operated by the National Meteorological Service at the Environmental Agency of the

Republic of Slovenia (ARSO), also has a comprehensive suite of meteorological sensors. The following parameters, recorded at 2 m a.g.l. and aggregated to hourly temporal resolution, were used in this study: air temperature (°C), relative humidity (%), wind speed (m s$^{-1}$), wind direction (°), and precipitation (mm h$^{-1}$) as well as standard air quality parameter (PM$_{10}$) (ARSO, 2017). To supplement these observations, total cloud



cover was obtained from the global data assimilation system (GDAS) model of NOAA/ARL (National Oceanic and Atmospheric Administration/Air Resources Laboratory).

In addition to the LjB air temperature measurements near the bottom of the Ljubljana basin (299 m a.s.l.), ARSO air temperature observations were also retrieved from a number of weather stations on the sidewalls of

the basin at a range of elevations from 268 to 1742 m. A full list of station names, acronyms and elevations is provided in Table 1, and their locations relative to LjB are shown in Figure 1c. To facilitate subsequent analyses, each station was assigned an "elevation category" (low, medium or high) according to its elevation relative to the bottom of the basin (Table 1).

The statistical analyses for this study were performed using the ggplot2 (Wickham, 2009) and ImputeTS

(Moritz, 2017) packages under R 3.3.1 (R Development Core Team, 2008). All reported times are local (LT = UTC + 1h) and the Northern Hemisphere seasonal convention has been adopted (i.e., winter: December–February; spring: March–May; summer: June–August; autumn: September–November).

**2.3 Identifying 'persistent temperature inversion' events**

**2.3.1 The radon-based method (RBM)**

Our method for identifying PTI events is based on continuous, near-surface, single-height observations of atmospheric radon concentrations. The principle of the method is outlined in the following two steps: (i) minimise or separate contributions to the observed radon variability occurring at the four main timescales (sub-diurnal, diurnal, synoptic and greater-than synoptic), and then (ii) compare the relative magnitudes of variability contributed by the diurnal and synoptic timescales. Since the separation and comparison of different timescale

contributions to radon concentration variability is performed separately for each season and year, changes in mean rates of radon emission between seasons or years do not adversely affect the efficacy of the technique.

Although the performance of the method is seasonally independent, the greatest number of PTI events in the SAB occurs in winter, so here we focus only on winter observations for the purpose of demonstrating the method. The processes involved in minimising or separating contributing timescales to the observed radon time

series in order to isolate the synoptic timescale contributions for further analysis (i.e., Step 1) are outlined below:

1.  **Minimising instrument noise and sub-diurnal radon variability:** since afternoon radon concentrations can be close to the instruments detection limit (even in winter; Figure 2a, black line),
instrument noise cannot be neglected. Furthermore, as outlined in Section 1 numerous factors contribute to natural variability in observed radon concentrations at sub-diurnal timescales. To minimise the impact of the above contributions on our estimation of diurnal minimum radon concentrations we started by performing a 3-hour running mean smoothing on the observed radon concentrations.

2.  **Characterisation of the synoptic variability ("fetch effect"):** mean summer and winter diurnal radon cycles at LjB are shown in Figure 3a. As detailed in Section 1, these cycles are characterized by peak concentrations near sunrise, when near-surface mixing is usually supressed, and minimum values in the mid-afternoon, when the ABL is deep (typically ≥ 1 km) and uniformly mixed. Depending on season



(specifically, the duration and intensity of solar radiation), the timing and duration of the diurnal maximum and minimum concentration periods may change by several hours (Figure 3a). For a given season, afternoon minimum radon concentrations are primarily influenced by the combined radon emissions along the air mass' recent (2−3 radon half-lives) fetch, which changes with the passage of

5 synoptic weather systems. Diurnal variability (driven by changes in mixing depth and the *local* radon flux) is superimposed *on top* of this fetch-driven radon variability. As an economical alternative to vertical radon gradient measurements (which require multiple *research-grade* detectors), the fetch-related influence (synoptic contribution) to radon concentrations observed at a single height can be *approximated* by linearly interpolating between successive afternoon minimum values (Chambers et

al., 2015; Podstawczyńska and Chambers, 2018). For the purpose of this investigation we derived the daily minimum radon concentrations over the year of observations by identifying the minimum smoothed concentration each afternoon between 14:00−19:00 LT. An example of this linearly-interpolated approximation of the air mass fetch contribution to the LjB radon concentrations is shown in Figure 2a (red line). It should be noted that the assumptions upon which this fetch-effect

approximation is based are invalid under severe weather conditions (e.g., during the passage of strong frontal systems, or rapid changes from terrestrial to oceanic fetch).

3. **Isolation of synoptic from longer timescale radon contributions:** variability on greater than synoptic timescales was evident in the radon time series, which needed to be minimised prior to attempting to

20 identify PTI events. The first step was to characterise typical synoptic timescale variability at LjB. We did this by performing a spectral analysis of the hourly LjB atmospheric pressure observations for each season (Figure 3b; winter example). Throughout winter and autumn our analysis indicated a synoptic timescale of 6 to 8 days, whereas in spring and summer the synoptic timescale was typically 7 to 11 days. We then calculated a running-minimum radon concentration using a window derived from the

25 spectral analysis of atmospheric pressure (7 day window for winter and autumn, and 11 days for spring and summer). Lastly, this running-minimum concentration was subtracted from the fetch-effect derived in (2) above, in order to isolate the synoptic time scale contributions to the radon observations (Figure 2b; red line "synoptic $C_{Rn}$").

Having isolated the synoptic scale contribution to the observed radon time series, the next steps to develop the PTI identification tool were to (i) identify periods when synoptic variability dominates diurnal variability, and (ii) decide on a minimum length of time that this needs to occur for the event to be considered a PTI.

The diurnal (mixing-related) contribution to the LjB radon time series was obtained by subtracting the advected (fetch-related) signal from the measured radon concentrations (see Figure 2a). For the purpose of this study we

chose the seasonal standard deviation of the diurnal radon contribution as a measure of the significance of diurnal variability each season. For our winter example, in Figure 2b we plotted the isolated synoptic radon signal $((C_{Rn})_{synoptic})$ and the standard deviation of the winter diurnal radon signal $((+\sigma_{DRn})_{winter})$. Numerous cases were evident when accumulation of radon due to synoptic processes $((C_{Rn})_{synoptic})$ exceeded $(+\sigma_{DRn})_{winter}$. At such times, synoptic controls on the near-surface radon observations are more significant than diurnal controls.

The final step is to set a duration threshold on these exceedances for an event to be classified as a PTI.



We consider a 'persistent temperature inversion' event to have occurred when the synoptic contribution to the radon signal exceeds the standard deviation of the diurnal contribution for that season for a period of at least 48 hours:

$$(C_{Rn})_{synoptic} > (+\sigma_{DRn})_{winter} \text{ for } \geq 48 \text{ hours} \qquad \qquad \dots (1)$$

Although only a winter example is presented in Figure 2b, the same approach was used for all seasons.

### 2.3.2 The pseudo-vertical temperature gradient method (TGM)

If distributed ground-based air temperature measurements are available over a wide range of representative elevations within a basin or valley it is possible to identify PTI events by analysing pseudo vertical temperature gradients between stations with significantly different elevations. This method, described by Largeron and Staquet, (2016a), relies upon two key assumptions: (1) horizontal homogeneity of air temperature between stations of similar elevations, and (2) the existence of significant vertical changes in air temperature between stations at different elevations. The validity of these assumptions can be assessed by calculating correlations between stations within designated elevation groups, and between stations across designated elevation groups. If the assumptions hold, temperature gradients between stations within the designated elevation groups can be calculated, and a threshold value determined to identify PTI events.

As detailed in Table 1, in the case of the Ljubljana basin, air temperature observations were available from eight automatic weather stations representative of three different elevation categories (low ~310 m; medium ~630 m; and high ~1320 m) for detecting PTI events. Firstly, all hourly temperature observations from the eight stations over the December 2016 to November 2017 period were separated by season. Next, seasonal correlation coefficients between each of the eight stations were calculated separately (Table 2). Air temperatures from the three low-elevation stations (Li, LjB and LLj) were strongly correlated for all seasons, despite a spatial separation of up to ~60 km (Figure 1c). Likewise, air temperatures of the two medium-elevation stations (S and T) were strongly correlated for all seasons. In the case of the high-elevation stations (PR, K and Kr), however, only moderate correlation coefficients were obtained (Table 2), indicating a degree of non-homogeneity of the horizontal air temperature field near the top of the basin.

Vertical temperature gradients between the low and medium elevation stations, and the medium and high elevation stations, were not as distinct in spring, summer or autumn as they were in winter, as indicated by strong correlation coefficients between temperatures from those elevations (Table 2). Thus, according to Whiteman et al., (2004), the actual vertical temperature profile of the basin's atmosphere can only be suitably approximated by pseudo-vertical temperature gradient measurements from the ground based stations over the whole year at night, when convective mixing is not active. Only in the colder months (winter and some of autumn; under low gradient wind conditions), can the pseudo-vertical temperature gradient be used effectively over the whole diurnal cycle.

Making the assumption of year-round horizontal homogeneity of the temperature field (Largeron and Staquet, 2016a), a relative measure of the degree of stability of the ABL can be made by calculating the vertical temperature gradient between stations of different elevation categories (e.g., high/medium, high/low, or medium/low), $(\Delta T/\Delta z)_i$, where the index $i$ refers to any pair of stations. To this end, following (Largeron and Staquet, 2016a), we calculated pseudo vertical temperature gradients for four station pairs. Of the 21 possible gradient combinations we sought to avoid medium/high gradient pairs (since the high stations were not



internally well correlated), and avoided the use of station S in the medium height category due to its comparatively high correlation with temperatures of the low stations. Other combinations were chosen to minimise spatial separation. The gradient pairs chosen for this study were as follows:

    i)     Li and K, $(\Delta T/\Delta z)_1$, 29 km apart, $\Delta z = 943$ m;
    ii)    LjB and T, $(\Delta T/\Delta z)_2$, 19 km apart, $\Delta z = 396$ m;
    iii)   LjB and PR, $(\Delta T/\Delta z)_3$, 35 km apart, $\Delta z = 720$ m;
    iv)   LLj and Kr, $(\Delta T/\Delta z)_4$, 23 km apart, $\Delta z = 1378$ m.

Following Largeron and Staquet, (2016a) we derived our basin stability measure (the pseudo-vertical temperature gradient) from the average of the four separate gradients:

$$\frac{\Delta T}{\Delta z}(t) = \frac{1}{4}\sum_{i=1}^{4}\left(\frac{\Delta T}{\Delta z}\right)_i(t) \qquad \qquad \ldots (2)$$

We note, however, that $\Delta T/\Delta z$ and the individual gradients $(\Delta T/\Delta z)_i$ were typically very similar:

$$\frac{\Delta T}{\Delta z}(t) \approx \left(\frac{\Delta T}{\Delta z}\right)_i(t), \text{ for } i=1,\ldots,4 \qquad \qquad \ldots (3)$$

In order to avoid the detection of successive strong nocturnal-only temperature inversions and isolate 'persistent temperature inversion' events, we applied a 24-hour running mean smoothing to pseudo-vertical temperature gradient $((\Delta T/\Delta z)_{24\text{-h}})$. Lastly, we calculated seasonal average values of $\Delta T/\Delta z$ to use as our seasonal thresholds for PTI event identification; a slight deviation from the approach of Largeron and Staquet, (2016a),

which was to use the "cold season" average (Nov−Dec).

A 'persistent temperature inversion' event is considered to have occurred when the $(\Delta T/\Delta z)_{24\text{-h}}$ value exceeds the seasonal average value for $\geq 48$ hours:

$(\Delta T/\Delta z)_{24\text{-h}} > (\Delta T/\Delta z)_{\text{season}}$ for $\geq 48$ hours.         $\ldots (4)$

As an example of this technique we present the winter event selection results in Figure 4. The same approach was used for all seasons.

**3 Results and Discussion**

**3.1 Data overview**

The time series of hourly average radon concentration at LjB revealed a seasonal cycle, characterised by a winter maximum and summer minimum (Figure 5a). When monthly averages of the separated synoptic and diurnal scale contributions to the radon observations are considered (Figure 5b) it is clear that the seasonal cycle

is mainly controlled by synoptic factors (changes in air mass fetch).

From October through December monthly mean wind speeds were typically low (Figure 5b), associated with frequent synoptic stagnation events. These periods resulted in higher radon concentrations (long "time over land"). In January and February, even though monthly average wind speeds increased, recent air mass fetch was predominantly continental, as indicated by dominant NE-E winds in seasonal wind roses (not shown), resulting

in moderate monthly mean radon concentrations. The lowest synoptic contributions to observed radon concentrations occurred in spring and summer. During these times the mean monthly wind speeds were higher



(strong mixing), and wind directions were often from the southwest, in the direction of the Gulf of Venice / Adriatic Sea, so terrestrial fetch for air masses was limited.

From March through September in Figure 5a the variability in radon concentration appears to be of considerably higher frequency than from October through February. This is further evidence of a change in dominant influences from diurnal to synoptic timescales, associated with changing day length and intensity of solar radiation within the Ljubljana basin.

### 3.2 Radon-based PTI event selection (RBM)

The radon-based method (RBM) identified six PTI events in the basin over the first whole year of measurements. Four events were detected in winter (*hereafter* $(E_i)_W$), and two in autumn $((E_i)_A)$. Dates, times and durations of these events are summarised numerically in Table 3, and graphically in Figure 2b (winter) and Figure 6b (autumn). The RBM detected no PTI events in spring and summer (see Figures 7 and 8).

According to the RBM approximately 20 % of winter experienced persistent strongly stable conditions, associated with anticyclonic synoptic conditions in the basin and surrounding regions. To characterize the meteorological conditions associated with a 'typical PTI event day' in winter, as determined by the RBM, we formed diurnal composites of some key meteorological quantities using hourly means from all 15 PTI event days in winter (Figure 9). As evident in the figure, winter PTI event days were mainly associated with modest diurnal temperature amplitudes (~5 °C on average), with morning minimums mostly between 0 to –4 °C. The diurnal amplitude of relative humidity was ~15 % on average (Figure 9b), with little variability in the extreme $(\mu+1\sigma)$ values (not shown). Wind speeds between sunset and sunrise were low and relatively consistent (on average 0.5–0.7 m s$^{-1}$), from the ESE (Figures 9d and 9e), with around 25 % cloud cover predicted and no measured rainfall; all consistent with anticyclonic synoptic conditions. The high variability in NOAA GDAS nocturnal cloud amount (Figure 9c) may be associated with fog that frequently forms under PTI conditions. It should be mentioned, there was no snow during the 2016−2017 winter, which would otherwise have led to stronger and longer-lasting persistent temperature inversion conditions. Results from a companion study to this one, still in preparation, indicate that snow cover in the following 2017–2018 winter reduced mean winter rates of radon emission by a factor of 2–3. The mean atmospheric pressure during PTI events was ~994 hPa, compared to ~986 hPa for non PTI conditions, confirming the prevalence of anticyclonic conditions.

By contrast, the RBM classified only 7 % of autumn as PTI events. Events $(E_1)_A$ and $(E_2)_A$ lasted four and two days, respectively, and both occurred in the second half of autumn. The composite diurnal meteorological conditions associated with the autumn PTI events are shown in Figure 10. Of particular note, wind speeds for the autumn PTI events were even lower than those in winter, typically 0.1–0.3 m s$^{-1}$ throughout the night, but also from the ESE. The mean diurnal amplitude of wind speed on autumn PTI days was ~0.5 m s$^{-1}$ compared to ~2.5 m s$^{-1}$ for non-PTI autumn days (Figure 10d). Between 00:00 and 05:00 LT near-surface temperature changed very slowly on autumn PTI nights (around –0.1 °C h$^{-1}$; Figure 10a) compared with around –0.4 °C h$^{-1}$ for non-PTI nights (Figure 10a). While some fog or cloud occurred on autumn PTI nights, no rainfall was recorded (Figures 10c and 10f).



### 3.3 Pseudo-vertical temperature gradient PTI event selection (TGM)

A total of 17 PTI events were detected throughout the measurement year by the TGM. Dates, times and duration of these events are summarised in Table 4. Five events were detected in winter (*hereafter* $(I_i)_W$), three in spring $((I_i)_{Sp})$, two in summer $((I_i)_S)$, and seven in autumn $((I_i)_A)$, as illustrated in Figures 4, 11, 12 and 13.

5 Approximately 30 % of winter days experienced persistently stable conditions according to the TGM. All of the TGM-classified events lasted from 3 to 8 days and were associated with strong inversion conditions (based on the mean vertical temperature gradients) with the exception of $(I_4)_W$, which was of lower intensity (Figure 4). Diurnal composites of air temperature, relative humidity, cloud cover, wind speed, wind direction, and precipitation for the winter TGM PTI events are shown in Figure 9. The mean diurnal temperature amplitude (6

10 °C) was 1 °C larger than that of the RBM events, and the diurnal amplitude of relative humidity (20 %) was 5 % larger than that of the RBM events, and nocturnal extreme relative humidity values were less consistent. Diurnal minimum wind speeds were slightly higher (0.6−0.7 m s$^{-1}$), and had a considerably larger diurnal amplitude than that of the RBM events. Of particular note, the TGM selected PTI days had more cloud cover (by 10−15 %) and experienced some rainfall (Figures 9c and 9f), which is very uncharacteristic of stable (typically

anticyclonic) conditions. It should be noted that many of the differences between RBM and TGM PTI events selected in winter can be attributed to the single event $(I_4)_W$, for which the mean temperature gradient indicated a lesser degree of stratification (Figure 4). Atmospheric pressure (not shown) was also elevated for all $(I_i)_W$ except $(I_4)_W$.

The highest number of PTI events identified by the TGM was in autumn (Figure 11). Around 32 % of this

season was classified as PTI conditions. The identified $(I_i)_A$ had durations of 2 to 12 days, with $(I_1)_A$, $(I_2)_A$, and $(I_7)_A$ being the shortest and with the least intense stability. The longest lasting event with the strongest stability was $(I_3)_A$. Diurnal composites of meteorological quantities corresponding to the autumn PTI events showed moderate amplitude temperature variability (5 to 15 °C), a 37 % amplitude change in relative humidity (with relatively consistent peak values between 01:00−06:00 LT, unlike spring and summer), and a 1.5 m s$^{-1}$

amplitude in wind speed (with a broad nocturnal minimum 0.2−0.3 m s$^{-1}$) (Figure 10a, 10b and 10d). Relatively high cloud cover amounts were associated with these events (around 40 %), with occasional rainfall recorded (Figures 10c and 10f).

In contrast to the RBM, the TGM also identified PTI events in spring (22 %) and summer (26 %) (Figures 12 and 13). Diurnal composites of meteorological conditions associated with these events are summarised in

Figures 14 and 15. Based on the spring temperature, relative humidity and wind speed diurnal composites, which all show gradual and relatively consistent changes throughout the night to brief (not broad) minimum/maximum values around 05:00−06:00 LT, followed by a rapid change after sunrise, the days identified here as PTI events actually correspond with days on which a strongly stable *nocturnal* boundary layer has formed, which is then broken up in the morning by strong convection, rather than *persistent* inversion

conditions. Further in support of this observation, only between 00:00 and 05:00 LT on the spring days identified by the TGM as PTI events did the wind direction switch to ESE, the expected direction of katabatic drainage at this site. The large mean diurnal amplitudes of temperature (~13 °C), relative humidity (~45 %), and wind speed (~2.5 m s$^{-1}$) in spring were not consistent with characteristics usually associated with PTI events. Similar characteristics were observed within the diurnal composite plots of summer PTI events identified by the

TGM, indicative of stable nocturnal conditions (though not as strong in summer) that rapidly become convective



after sunrise, rather than persisting as stable events throughout the day. Further evidence that the TGM was even less selective of nocturnally stable conditions is the frequent rainfall events observed on the nights in question (Figure 15f).

### 3.4 Comparison of the radon-based and temperature gradient methods

Importantly, all winter and autumn PTI events identified by the RBM were also identified by the TGM (Table 3 and 4). However, the TGM classified a number of other periods in winter and autumn as PTI events, and also considered that the $(I_3)_A$ was longer lasting than predicted by the RBM. Furthermore, the TGM appears to have falsely identified numerous days in spring and in summer as PTI events, whereas there is clear evidence in the meteorological composites that strong daytime convection in these cases was active between periods of strong

nocturnal stability.

By definition, PTI events should not exhibit strong diurnal cycles of temperature and relative humidity, should have consistently light-to-calm nocturnal wind speeds and, since little daytime convective mixing is expected, daytime wind speeds should also be relatively light. Furthermore, since they usually occur under anticyclonic synoptic conditions, when regional subsidence prevails, rainfall is not expected.

Comparing Figure 9 with Figure 10, it is apparent that diurnal cycle amplitudes of temperature, relative humidity and wind speed for the RBM PTI events are consistently lower than for the TGM selected PTI events. Hourly average wind speeds for the RBM selected PTI events are also low ($\mu+1\sigma \leq 1-1.2$ m s$^{-1}$) across the whole diurnal cycle for composite winter and autumn events, with broad ($\geq$ 12-hour periods) of consistently low wind speeds (0.2–0.5 m s$^{-1}$). By comparison, PTI events identified by the TGM had shorter nocturnal periods of

lowest wind speed (5−8 hours), some mean hourly wind speeds >1 m s$^{-1}$, with daytime $\mu+1\sigma$ values of 2–3.2 m s$^{-1}$ (not shown). Lastly, in contrast to results from the TGM, from prior to sunset until several hours after sunrise, no rainfall was recorded at all under PTI conditions identified by the RBM in either winter or autumn.

In summary, while no PTI events identified by the RBM in winter or autumn were missed by the TGM, it is clear that the TGM is less selective of truly persistently stable conditions than the RBM. This shortcoming is

25 likely attributable to the degree of validity of the assumptions on which the TGM is based for the Ljubljana Basin. In spring and summer (Figures 14 and 15) some assumptions of the TGM are less valid (i.e., less distinction between temperatures at different elevations), causing the method to produce misleading results. Large temperature gradients on very stable nights' push daily mean values well above the seasonal mean. For consecutive strongly stable nights, under anticyclonic conditions, this is incorrectly interpreted as a PTI event by

30 the TGM.

### 3.5 Advantages and limitations of radon-based method (RBM)

The RBM relies upon accurate characterisation of afternoon minimum radon concentrations. In this regard, measurement quality and time series analysis are important. The lower limit of detection of commercial, portable radon detectors such as the AlphaGUARD (LLD 2−3 Bq m$^{-3}$) is two orders of magnitude poorer than

35 that of research-grade two-filter radon detectors (LLD ~0.02 Bq m$^{-3}$; Chambers et al., 2014). Compared with sub-diurnal variability this necessitates smoothing of radon observations if using a commercial instrument, the extent of which is likely to be application/site specific. An alternative, but more expensive, solution would be to make vertical radon gradient measurements using research-grade detectors. This would ensure the most



accurate, and simple, separation of diurnal (mixing related) and synoptic (fetch related) contributions to observed radon concentrations.

The amount of radon that accumulates over a night or a given synoptic PTI event, is directly related to the seasonal average radon flux. By performing the RBM separately for each season and year, long-term (i.e., seasonal mean) changes of the terrestrial radon flux from one season or year to the next (as may be driven by changes in mean soil moisture, snow cover or soil freezing) do not affect the comparative seasonal efficacy of the technique, making it seasonally independent. However, large spatial variability in the radon source function (as experienced between land and open water bodies), have the potential to prevent this technique from accurately identifying PTI events, but only for study regions located on (or close to, i.e., $\leq 10$ km) the coast. For non-coastal regions, the longevity of radon's parent ($^{226}$Ra, half-life 1600 years) ensures that the technique can be applied consistently over long periods, and – when enough PTI events have been observed within a given season – provide a means of objectively assessing their severity.

Compared to the pseudo-vertical temperature gradient method, which requires input from multiple ground-based weather stations (Whiteman et al., 2004), single height radon monitoring, from a single monitoring station near the bottom of the basin/valley, clearly provides a simple, economical and low maintenance means by which to identify 'persistent temperature inversion' events with a substantially higher degree of success than some contemporary meteorological approaches.

### 3.6 Advantages and limitations of the pseudo-vertical temperature gradient method (TGM)

For the Ljubljana Basin the TGM was found to not always be a reliable means of detecting PTI events. This was mainly due to limited validity of the necessary assumptions. The assumption of a horizontally homogeneous temperature field was often invalid in the case of the high elevation sites, where a variety of mesoscale processes, including drainage flows, nocturnal jets, and intermittent turbulence (Whiteman et al., 2004; Williams et al., 2013) limit the ability to derive representative vertical temperature gradients from spatially separated locations at different elevations. Furthermore, the assumption of distinct differences between temperatures of the different elevation groups based on daily means was typically only valid in winter (Table 2). Consequently, for most of the year the TGM was only able to reliably identify stable conditions in the boundary layer at night, when buoyancy or gravitational effects tend to produce horizontal isotherms (Whiteman et al., 2004). However, pseudo-vertical temperature profiles *are* useful for identifying extended periods of strongly stable conditions based on 24-hour data in winter (Largeron and Staquet, 2016a). According to Whiteman et al. (2004), the TGM tends to be more representative of PTI events when the study region has snow cover. A uniform cover of snow reduces the temperature contrasts usually associated with radiation receipt on different aspect slopes and different types of ground cover, helping to maintain stability over the diurnal cycle (Whiteman et al., 2004). The reduced accuracy of the technique in the SAB during the 2016−2017 winter season may be attributable in part to a lack of snow cover that winter.

The number and location of ground-based weather stations are important factors of the TGM; stations should be placed in all directions (north, east, south and west). The Ljubljana Basin automatic weather station network lacked stations in the south, which may have contributed to the poor performance of the technique. Large open regions of complex topography, more exposed to synoptic wind influences, are not well suited to the TGM (Whiteman et al., 2004).



Overall, the inability of the TGM to perform equally well in all seasons, and its lack of appropriate selectivity of truly persistent stable conditions (regarding event number, duration and strength) in winter and autumn compared to the RBM, were the most serious limitations.

**3.7 Urban air quality during 'persistent temperature inversion' events**

In some regions of Europe with complex terrain, urban air quality is strongly influenced by PTI events. For example, of 29 separate air pollution events in the Eordea mountain basin of north eastern Greece analysed by Triantafyllou, (2001), more than half occurred under PTI conditions. Other European regions in which of cold season PTI events have been studied in complex topographical locations include: Athens, Greece (Kassomenos and Koletsis, 2005); Krakow, Poland (Bielec-Bąkowska et al., 2011); Grenoble (Largeron and Staquet, 2016a)

and Chamonix-Mont-Blanc (Chemel et al., 2016), France; Ljubljana (Petkovšek, 1978, 1985, 1992; Rakovec et al., 2002) and Ajdovščina (Longlong et al., 2016), Slovenia; the Po Valley region, Italy (Marcazzan et al., 2001). The pollution events in each of these studies were characterized by stable stratification and subsidence, suppressing vertical mixing while the confining topography and weak-to-calm gradient winds prevented advection and favoured air stagnation (Kukkonen et al., 2005). Under such circumstances the accumulation of

primary pollutants and formation of secondary pollutants not only poses adverse pulmonary and cardio-vascular health effects, but also greatly reduces the visibility (Largeron and Staquet, 2016a, 2016b; Reddy et al., 1995; Silcox et al., 2012; Whiteman et al., 2014), and can lead to hazardous episodes of persistent freezing rain, drizzle, or fog (Petkovšek, 1974).

In order to better compare the suitability of the RBM and TGM methods of PTI identification for air quality

assessment of subalpine basin cities, hourly observations of $PM_{10}$ in Ljubljana city were analysed. The diurnal and seasonal variability of $PM_{10}$ in Ljubljana are shown in Figure 16 for all conditions (seasonal mean), as well as for PTI conditions as determined by the RBM and TGM.

Overall, the annual mean $PM_{10}$ concentration for Ljubljana was 28 μg m$^{-3}$, and the average daily-maximum $PM_{10}$ concentration for winter was 53 μg m$^{-3}$, which exceeded the WHO and EEA standard of maximum daily

concentration 50 μg m$^{-3}$ (EEA, 2016, 2017; Ivančič and Vončina, 2015).

Compared with seasonal mean values, there was a significant increase of $PM_{10}$ concentrations under cold season PTI conditions, as a consequence of poor atmospheric mixing over the whole diurnal cycle (Figures 16a and 16b). Compared to the RBM, the TGM underestimated peak hourly $PM_{10}$ concentrations of PTI events by 13 and 11 μg m$^{-3}$, in winter and autumn, respectively. In winter $PM_{10}$ concentrations for PTI events were

30 underestimated across the entire diurnal cycle by the TGM, and for around 18 hours of the composite day in autumn. These differences would lead to significantly different levels of daily-mean exposure for residents. The amplitude of the diurnal cycle of $PM_{10}$ for TGM PTI events was 4 μg m$^{-3}$ larger than that of RBM events; attributable to more mixing at night and slightly stronger daytime atmospheric mixing in the case of TGM PTI events (Figures 16a and 16b).

In the warmer months, neither the daily mean $PM_{10}$ values for non-PTI or TGM PTI periods exceeded guideline values (Figures 16c and 16d). Diurnal cycles of $PM_{10}$ for TGM PTI events at these times characterized by higher concentrations in the morning (06:00–08:00 LT) and evening (18:00–20:00 LT) rush hours. In the afternoons when the ABL was deepest, concentrations of $PM_{10}$ were the lowest due to typical of diurnally-changing as opposed to persistent atmospheric mixing conditions.



The seasonal inconsistency of the TGM for PTI identification, in addition to its lack of selectivity of persistently stable conditions in the cold seasons, render it an unsuitable tool for air quality management purposes in subalpine basin environments.

Our findings, based on the RBM of PTI event identification, indicated that PTI events span 27 % of the cold
season (winter and autumn) in the subalpine Ljubljana Basin, making it one of the critical European regions with regard to urban air quality issues. Due to domestic heating requirements, the increased use of vehicles, and the production of energy from the thermal power plant, pollutant emissions drastically increase in the cold season. Adverse dispersion conditions as a result of frequent cold season PTI events favour the accumulation of pollutants, leading to severe health issues. For these reasons, improving knowledge about the frequency,
duration, intensity and seasonal variability of PTI events is required for decision makers when considering the need for emission mitigation strategies.

The technique developed in this study adds another dimension to contemporary radon-based methods of characterising the atmospheric mixing state (Chambers et al., 2015, 2018; Williams et al., 2016) that were developed for less complex topographic regions. Under PTI conditions contributions to radon concentration
variability on synoptic timescales can dominate variability on diurnal timescales, thereby violating a fundamental assumption of the approach employed by Chambers et al. (2018). Consequently, nocturnal periods under PTI conditions would be misclassified below their actual degree of atmospheric stability. Applying the RBM of PTI classification to observations in a subalpine basin setting after first classifying the data according to the scheme of Chambers et al. (2018) would enable clear identification of the misclassified events for separate
analysis. The combined application of these two approaches using a complete suite of criteria pollutants in Ljubljana city will be the subject of a separate investigation.

## 4 Conclusions

The aim of this study was to determine a reliable and seasonally-independent method to identify 'persistent temperature inversion' events in the subalpine basin setting of Ljubljana, based on observations over the period
December 2016−November 2017. To this end, two methods were compared: (i) a new radon-based method developed and tested in this study, and (ii) a previously published pseudo-vertical temperature gradient method. Development of the radon based method required an understanding of the various timescales contributing to variability in the 1-year hourly radon time series (sub-diurnal, diurnal, synoptic and greater than synoptic). The method is applied in four steps: (i) minimise sub-diurnal radon variability, (ii) identification of the fetch-related
(synoptic scale) radon variability, (iii) isolating the synoptic-scale from larger timescale variability, and (iv) identification of 'persistent temperature inversion' events using the seasonal standard deviation of diurnal timescale radon variability as an event threshold with a ≥ 48-hour persistence requirement. The method was applied separately for each season in order to account for seasonal changes in the radon source function, day length and atmospheric mixing.

The existing pseudo-vertical temperature gradient method to identify PTI events was applied using 1-year of measurements from eight automatic weather stations, within three distinct elevation categories, distributed around the base and on the sidewalls of the basin. The technique considers daily-mean temperature gradients between stations in different elevation categories. The applicability of the technique is contingent upon two key





assumptions: (i) horizontal homogeneity of air temperature between stations within the same elevation category, and (ii) clear and consistent distinction of temperature between stations in different elevation categories.

Over the year, the radon-based method identified six PTI events (4 in winter, 2 in autumn). No PTI events were identified in spring or summer. By comparison, the pseudo-vertical temperature gradient method identified 17

PTI events (winter 5; spring 3; summer 2; and autumn 7). All 6 events identified by the radon-based method were also identified by the pseudo-vertical temperature gradient method. Each of the events identified by both methods were associated with persistent strongly stable conditions; light-to-calm winds (typically 0.2−0.6 m s$^{-1}$), modest amplitudes of diurnal temperature and relative humidity cycles, relatively consistent nocturnal conditions, and no precipitation. On the other hand, the additional 11 events detected by the pseudo-vertical

temperature gradient method were characterised by somewhat mixed meteorological conditions: larger amplitude diurnal cycles of temperature, humidity and wind speed, daytime wind speeds of up to 4 m s$^{-1}$, and frequent nocturnal rainfall. The comparative suitability of the RBM and TGM of PTI event identification in air quality studies was done using hourly $PM_{10}$ observations. Peak hourly-mean $PM_{10}$ concentrations of TGM PTI events was 13 (11) μg m$^{-3}$ lower than that of RBM PTI events in winter (autumn). Only peak hourly-mean

values of RBM PTI events exceeded 100 μg m$^{-3}$, the upper threshold of low-level short-term $PM_{10}$ exposure according to World Health Organisation guidelines.

The RBM was found to be more reliable and seasonally-independent than the TGM. The inaccuracy and inconsistency of the TGM was thought to be primarily attributable to the requisite assumptions, which were not consistently met throughout the year. We demonstrated that the RBM is a convenient, simple and economical

means of consistently identifying PTI events in complex topographies (e.g., basins or valleys) regardless of season. The consistency and accuracy of this method has the potential to significantly increase the understanding of meteorological processes resulting in air pollution episodes. Once a representative number of PTI events have been observed within each season, this technique also has the potential to objectively assess the relative strength of PTI events. As such, the RBM has the potential to become a key tool in urban centres for air

quality management and assessing public health risks.

**Acknowledgements**

The authors acknowledge the financial support from the Slovenian Research Agency (research core funding No. P1-0143) and from the Public Scholarship, Development, Disability and Maintenance Fund of the Republic of Slovenia (Contract no. 11011- 44/2016-18). The authors also gratefully acknowledge the Environmental Agency

of the Republic of Slovenia for the permission provided and the assistance by installing the weather instrument shelter for radon monitoring on their land and for providing the meteorological data.

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



**Tables:**

**Table 1. Ground-based automatic weather station name, acronym, position, elevation and elevation category.**

| Station Name | Acronym | Latitude (°N) | Longitude (°E) | Elevation (m) | Elevation category |
|---|---|---|---|---|---|
| Litija | Li | 46°06' | 14°81' | 268 m | low |
| Ljubljana-Bežigrad | LjB | 46°07' | 14°52' | 299 m | low |
| Letališče Jožeta Pučnika Ljubljana | LLj | 46°21' | 14°47' | 364 m | low |
| Sevno | S | 45°98' | 14°92' | 556 m | medium |
| Topol | T | 46°09' | 14°37' | 695 m | medium |
| Pasja Ravan | PR | 46°09' | 14°22' | 1019 m | high |
| Kum | K | 46°08' | 15°07' | 1211 m | high |
| Krvavec | Kr | 46°29' | 14°53' | 1742 m | high |



**Table 2.** Air temperature correlation coefficients between each of the eight ground-based automatic weather stations in spring, summer, autumn and winter.

**Winter**

|      | Li | LjB | LLj | S | T | PR | K | Kr |
|------|----|-----|-----|---|---|----|---|----|
| Li   | 1 | 0.96 | 0.94 | 0.81 | 0.74 | 0.59 | 0.47 | 0.27 |
| LjB  |   | 1 | 0.95 | 0.84 | 0.78 | 0.59 | 0.47 | 0.26 |
| LLj  |   |   | 1 | 0.80 | 0.75 | 0.59 | 0.46 | 0.24 |
| S    |   |   |   | 1 | 0.96 | 0.80 | 0.69 | 0.46 |
| T    |   |   |   |   | 1 | 0.85 | 0.74 | 0.52 |
| PR   |   |   |   |   |   | 1 | 0.84 | 0.75 |
| K    |   |   |   |   |   |   | 1 | 0.75 |
| Kr   |   |   |   |   |   |   |   | 1 |

**Spring**

|      | Li | LjB | LLj | S | T | PR | K | Kr |
|------|----|-----|-----|---|---|----|---|----|
| Li   | 1 | 0.97 | 0.97 | 0.89 | 0.87 | 0.87 | 0.83 | 0.74 |
| LjB  |   | 1 | 0.96 | 0.91 | 0.91 | 0.80 | 0.86 | 0.80 |
| LLj  |   |   | 1 | 0.88 | 0.87 | 0.88 | 0.83 | 0.80 |
| S    |   |   |   | 1 | 0.97 | 0.96 | 0.96 | 0.90 |
| T    |   |   |   |   | 1 | 0.98 | 0.96 | 0.92 |
| PR   |   |   |   |   |   | 1 | 0.76 | 0.71 |
| K    |   |   |   |   |   |   | 1 | 0.94 |
| Kr   |   |   |   |   |   |   |   | 1 |

**Summer**

|      | Li | LjB | LLj | S | T | PR | K | Kr |
|------|----|-----|-----|---|---|----|---|----|
| Li   | 1 | 0.95 | 0.97 | 0.87 | 0.85 | 0.85 | 0.81 | 0.66 |
| LjB  |   | 1 | 0.96 | 0.91 | 0.91 | 0.98 | 0.83 | 0.72 |
| LLj  |   |   | 1 | 0.89 | 0.89 | 0.98 | 0.83 | 0.68 |
| S    |   |   |   | 1 | 0.97 | 0.96 | 0.96 | 0.90 |
| T    |   |   |   |   | 1 | 0.98 | 0.96 | 0.92 |
| PR   |   |   |   |   |   | 1 | 0.76 | 0.71 |
| K    |   |   |   |   |   |   | 1 | 0.94 |
| Kr   |   |   |   |   |   |   |   | 1 |

**Autumn**

|      | Li | LjB | LLj | S | T | PR | K | Kr |
|------|----|-----|-----|---|---|----|---|----|
| Li   | 1 | 0.97 | 0.95 | 0.88 | 0.85 | 0.82 | 0.80 | 0.70 |
| LjB  |   | 1 | 0.94 | 0.90 | 0.88 | 0.84 | 0.81 | 0.72 |
| LLj  |   |   | 1 | 0.85 | 0.83 | 0.82 | 0.78 | 0.70 |
| S    |   |   |   | 1 | 0.98 | 0.96 | 0.94 | 0.85 |
| T    |   |   |   |   | 1 | 0.97 | 0.95 | 0.86 |
| PR   |   |   |   |   |   | 1 | 0.77 | 0.78 |
| K    |   |   |   |   |   |   | 1 | 0.73 |
| Kr   |   |   |   |   |   |   |   | 1 |



**Table 3.** 'Persistent temperature inversion' events ($E_i$) by season (winter=W, spring=Sp, summer=S, autumn=A) detected by the radon-based method (RBM), including time period and duration.

| Event | Persistent inversion period | Duration |
|---|---|---|
| $(E_1)_W$ | 17/12/2016−19/12/2016 | 59 h (2.5 d) |
| $(E_2)_W$ | 23/12/2016−27/12/2016 | 101 h (4.2 d) |
| $(E_3)_W$ | 30/12/2016−02/01/2017 | 72 h (3 d) |
| $(E_4)_W$ | 28/01/2017−02/02/2017 | 131 h (5.5) |
| $(E_1)_A$ | 14/10/2017−18/10/2017 | 96 h (4) |
| $(E_2)_A$ | 19/11/2017−21/11/2017 | 48 h (2 d) |

5   **Table 4.** 'Persistent temperature inversion' events ($I_i$) by season (winter=W, spring=Sp, summer=S, autumn=A) detected by pseudo-vertical temperature gradient method (TGM), including time period and duration. *Note: the shaded events were also detected by the radon-based method (RBM).*

| Event | Persistent inversion period | Duration |
|---|---|---|
| $(I_1)_W$ | 17/12/2016−19/12/2016 | 62 h (2.6 d) |
| $(I_2)_W$ | 20/12/2016−27/12/2016 | 184 h (7.6 d) |
| $(I_3)_W$ | 29/12/2016−02/01/2017 | 93 h (3.8 d) |
| $(I_4)_W$ | 20/01/2017−24/01/2017 | 89 h (3.7) |
| $(I_5)_W$ | 27/01/2017−02/02/2017 | 152 h (6.3 d) |
| $(I_1)_{Sp}$ | 13/03/2017−21/03/2017 | 191 h (8 d) |
| $(I_2)_{Sp}$ | 25/03/2017−04/04/2017 | 246 h (10.3 d) |
| $(I_3)_{Sp}$ | 25/03/2017−04/04/2017 | 48 h (2 d) |
| $(I_1)_S$ | 31/07/2017−16/08/2017 | 386 h (16 d) |
| $(I_2)_S$ | 20/08/2017−27/08/2017 | 186 h (7.8 d) |
| $(I_1)_A$ | 01/10/2017−03/10/2017 | 51 h (2.1) |
| $(I_2)_A$ | 07/10/2017−09/10/2017 | 59 h (2.5 d) |
| $(I_3)_A$ | 10/10/2017−22/10/2017 | 277 h (11.5 d) |
| $(I_4)_A$ | 24/10/2017−29/10/2017 | 117 h (4.9 d) |
| $(I_5)_A$ | 30/10/2017−02/11/2017 | 79 h (3.3 d) |
| $(I_6)_A$ | 19/11/2017−21/11/2017 | 65 h (2.7 d) |
| $(I_7)_A$ | 26/11/2017−28/11/2017 | 50 h (2.1 d) |





**Figures:**

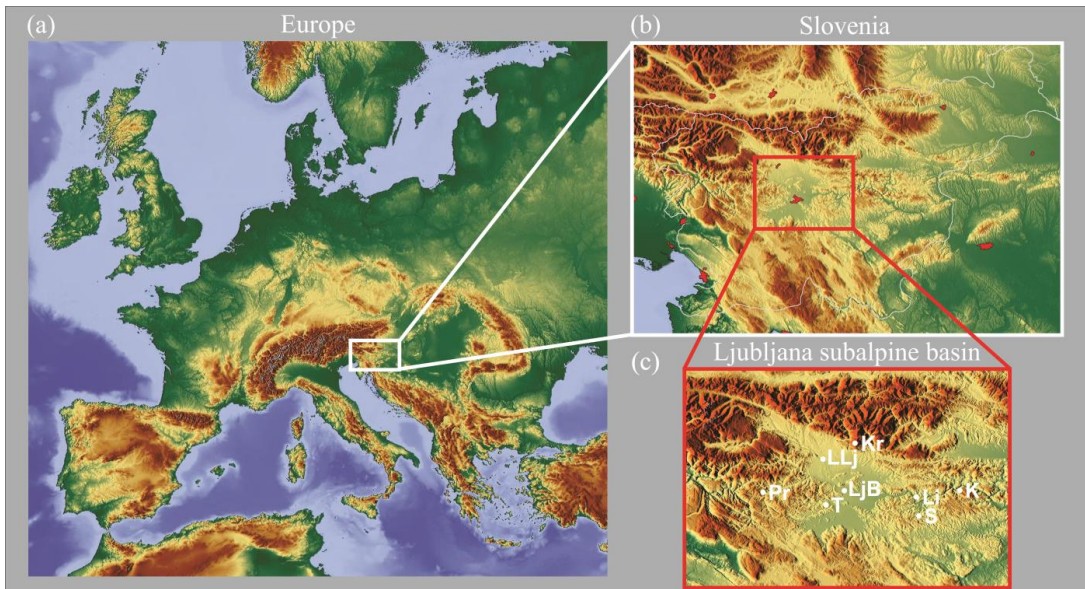

**Figure 1: (a) Location of Slovenia within Europe, (b) topographic map of Slovenia with the location of Ljubljana indicated, and (c) topographic map of the Ljubljana Basin, with the locations of automatic weather stations indicated.**





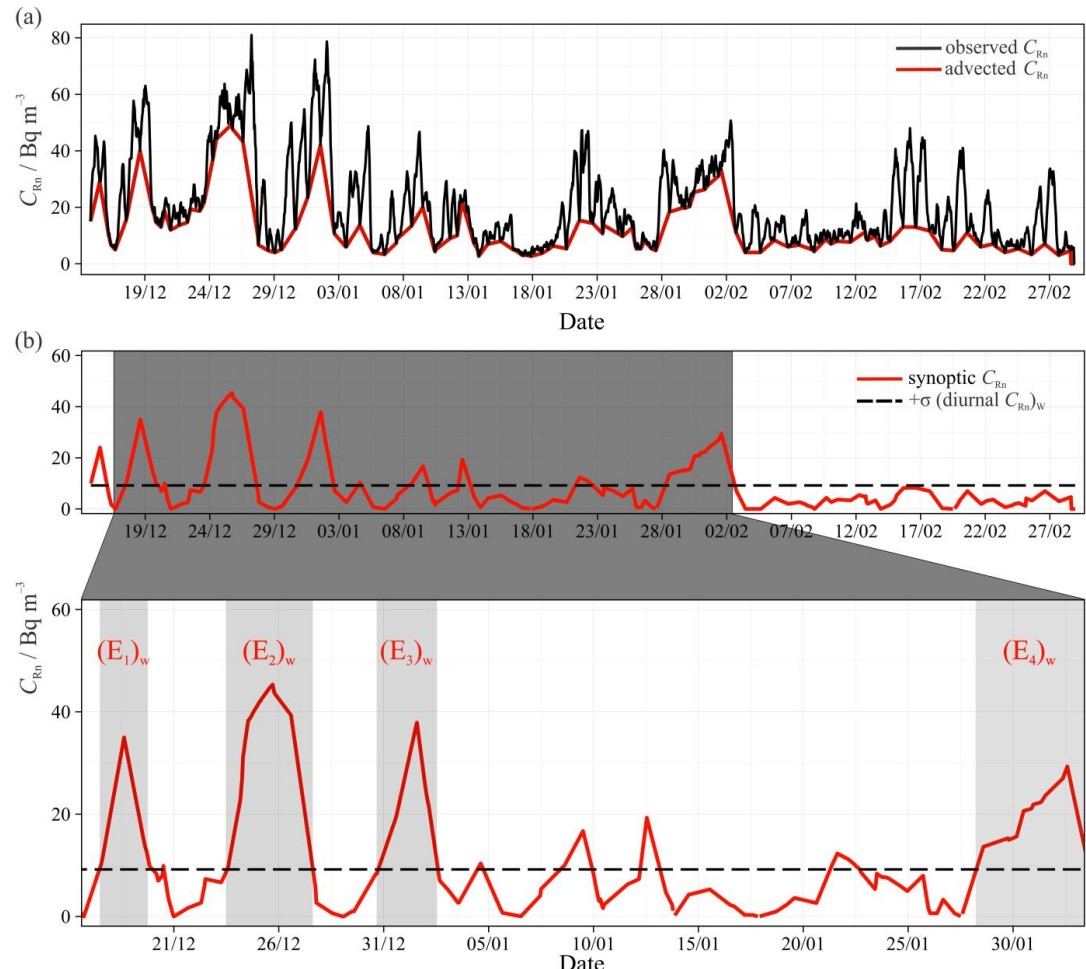

**Figure 2: (a) Hourly mean** *observed* **radon concentrations with the** *advected* **contribution indicated, and (b) isolated** *synoptic* **timescale radon contribution in winter with the standard deviation of the** *diurnal* **contribution indicated. Four 'persistent temperature inversion' events** $(E_i)_W$ **detected by the RBM are shown in the shaded area and enlarged in the breakout panel beneath; see text in section 2.3.1 for details.**

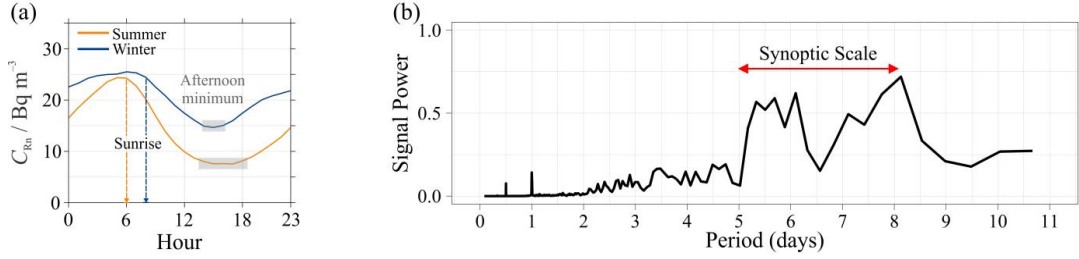

**Figure 3: (a) Summer and winter diurnal composite hourly radon concentration, and (b) spectral analysis of the LjB hourly atmospheric pressure data for winter.**





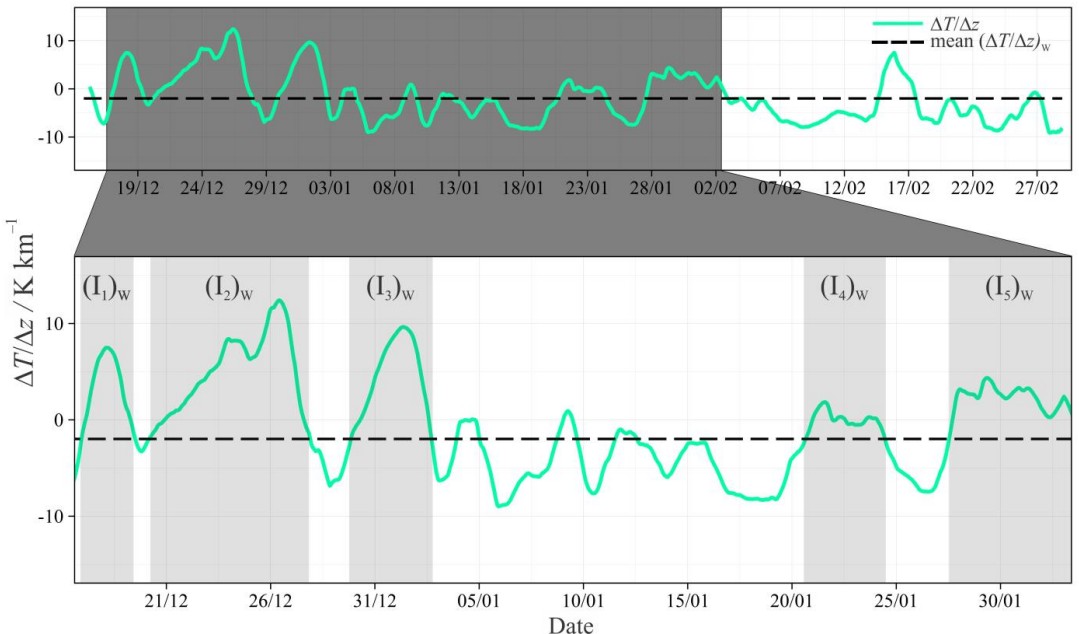

**Figure 4: 24-h running mean pseudo-vertical temperature gradient in winter. Five 'persistent temperature inversion' events $(I_i)_W$ detected by the TGM are shown in the shaded area and enlarged in the breakout panel beneath; see text in section 2.3.2 for details.**

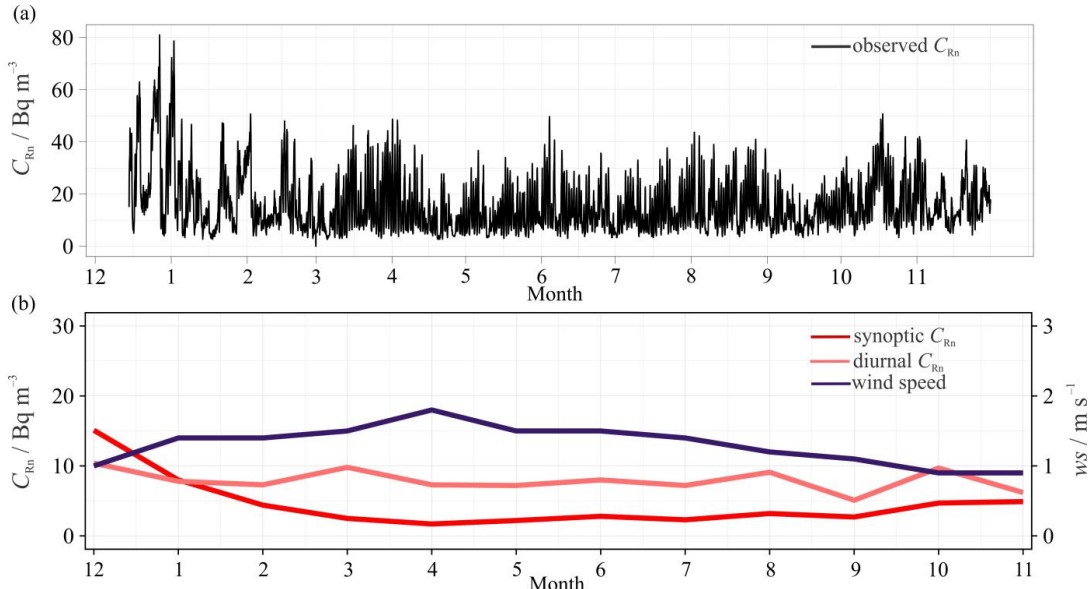

5    **Figure 5: (a) One year (Dec−2016 to Nov−2017) of hourly radon concentration, and (b) monthly average of synoptic radon, diurnal mixing of radon, and wind speed.**




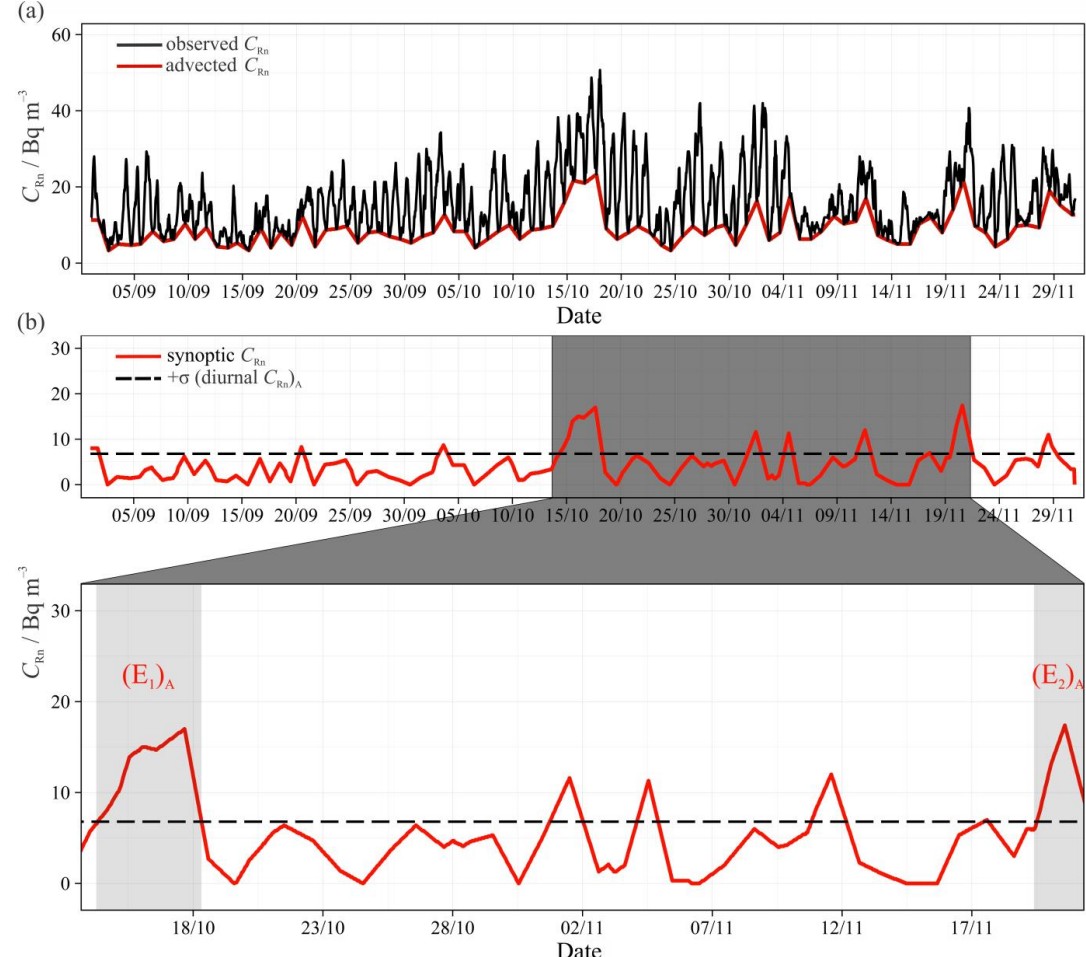

**Figure 6: (a) Hourly mean *observed* radon concentrations with the *advected* contribution indicated, and (b) isolated *synoptic* timescale radon contribution in autumn with the standard deviation of the *diurnal* contribution indicated. Two 'persistent temperature inversion' events (E$_i$)$_A$ detected by the RBM are shown in the shaded area and enlarged in the breakout panel beneath; see text in section 2.3.1 for details.**





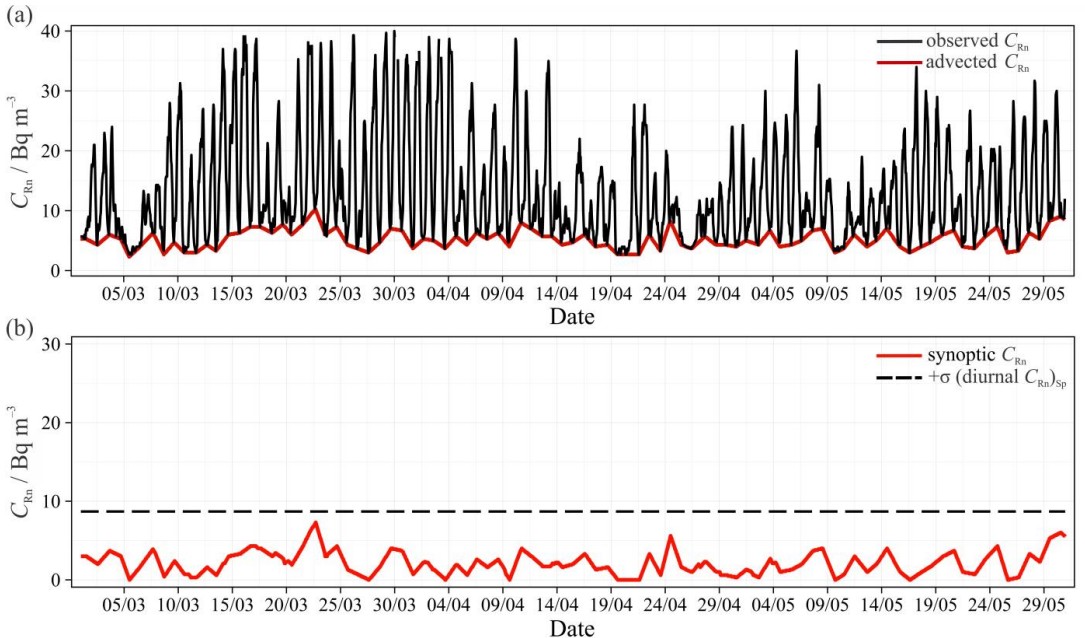

**Figure 7: (a)** Hourly mean *observed* radon concentrations with the *advected* contribution indicated, and **(b)** isolated *synoptic* timescale radon contribution in spring with the standard deviation of the *diurnal* contribution indicated. No 'persistent temperature inversion' events were detected by the RBM this season.

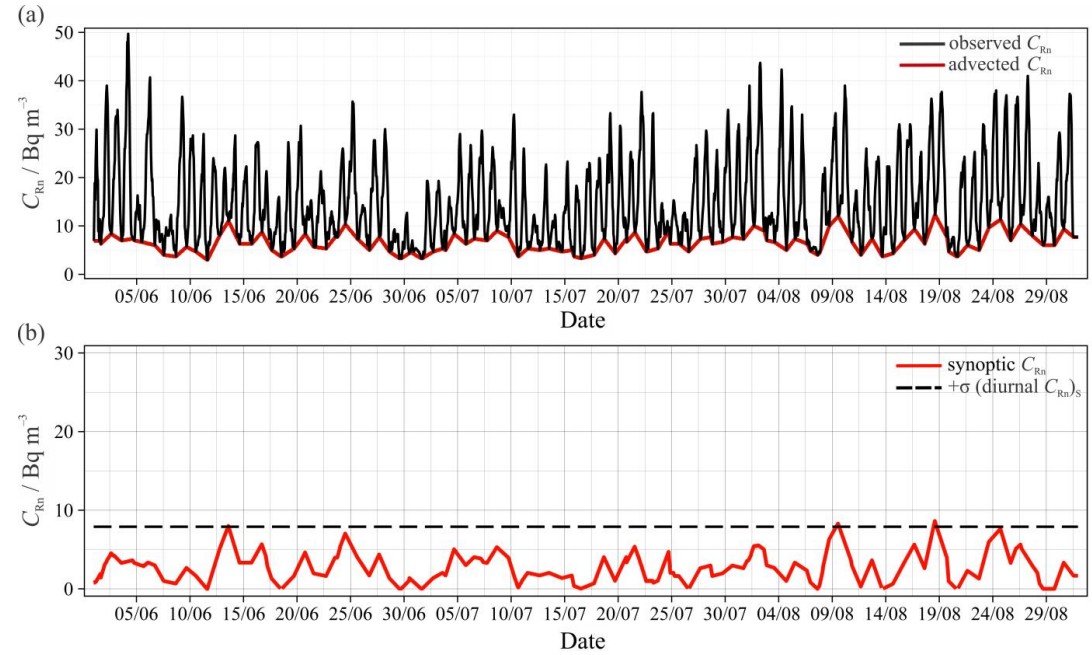

**Figure 8: (a)** Hourly mean *observed* radon concentrations with the *advected* contribution indicated, and **(b)** isolated *synoptic* timescale radon contribution in summer with the standard deviation of the *diurnal* contribution indicated. No 'persistent temperature inversion' events were detected by the RBM this season.

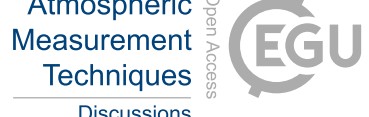



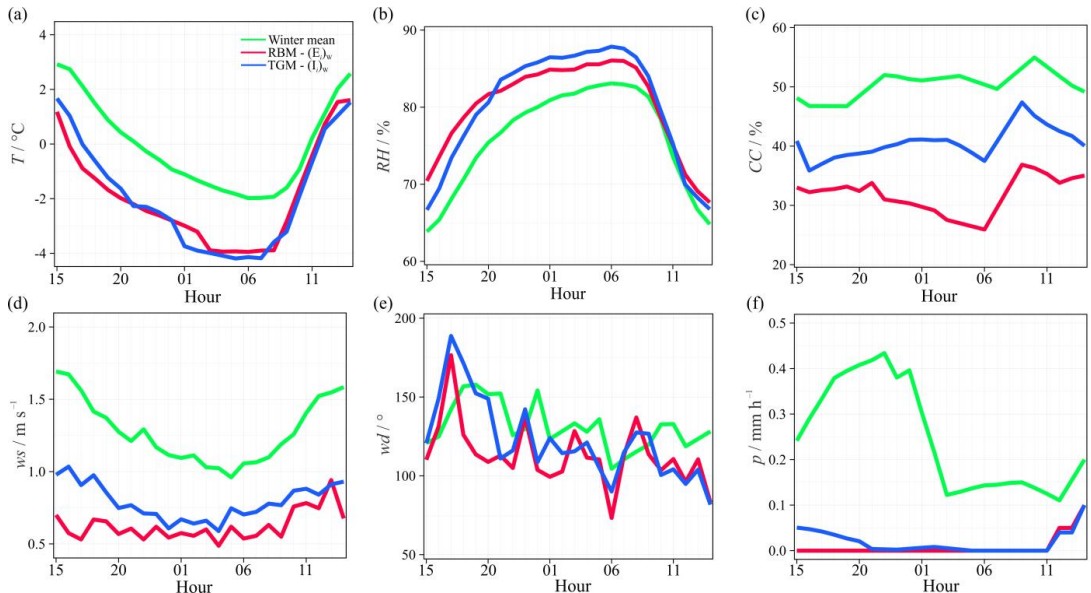

**Figure 9:** Diurnal composites, based on hourly means, of: (a) air temperature, (b) relative humidity, (c) cloud cover, (d) wind speed, (e) wind direction, and (f) precipitation, for winter mean, and 'persistent temperature inversion' events detected in winter by the RBM and TGM.

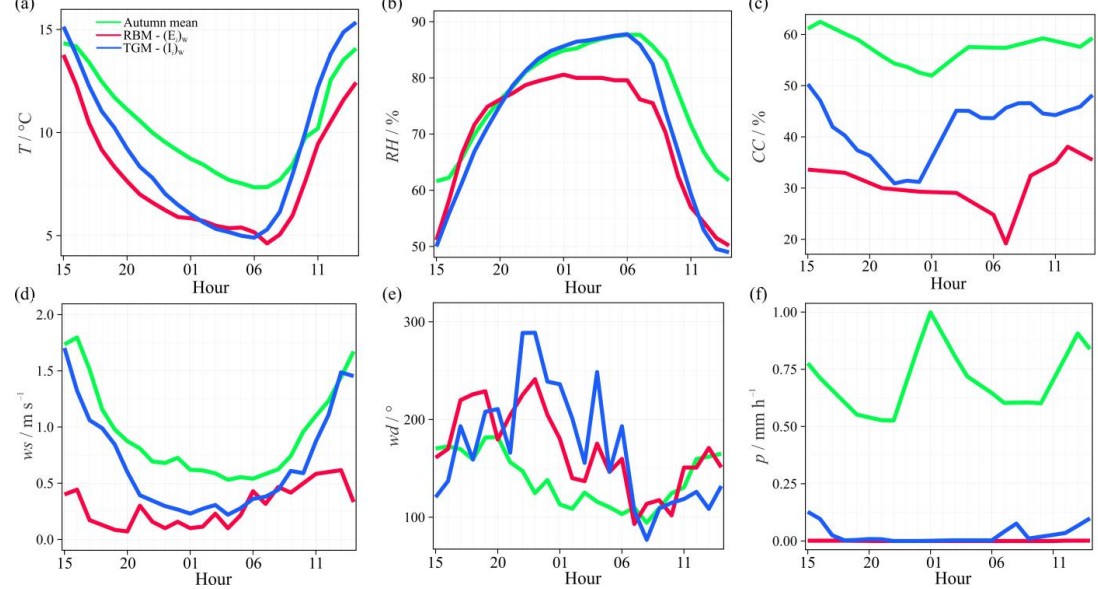

**Figure 10:** Diurnal composites, based on hourly means, of: (a) air temperature, (b) relative humidity, (c) cloud cover, (d) wind speed, (e) wind direction, and (f) precipitation, for autumn mean, and 'persistent temperature inversion' events detected in autumn by the RBM and TGM.




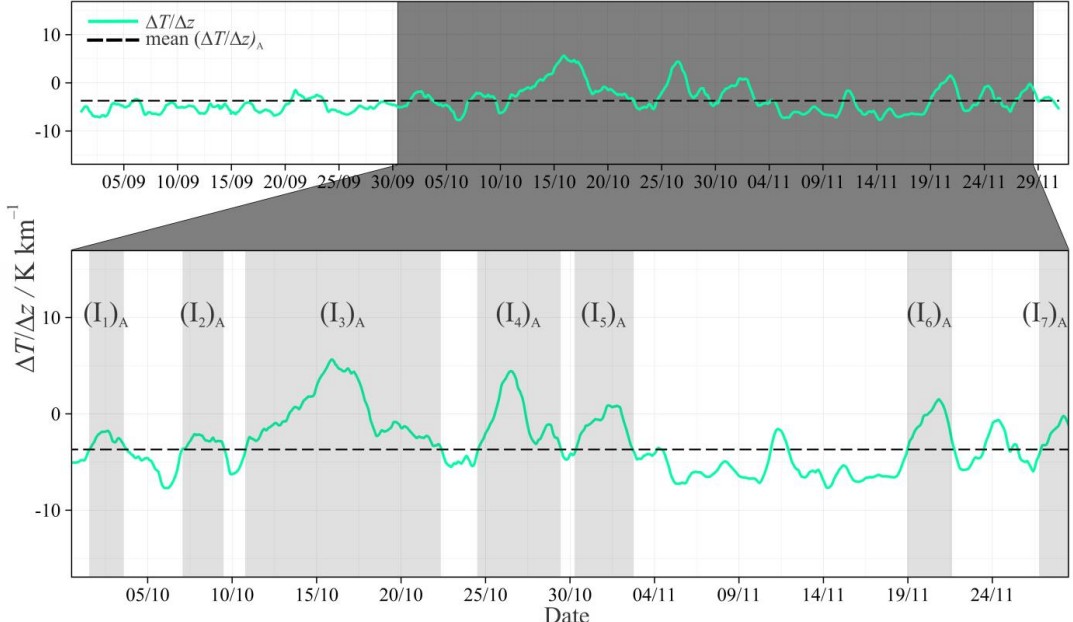

**Figure 11: 24-h running mean of the pseudo-vertical temperature gradient in autumn; seven 'persistent temperature inversion' events $(I_i)_A$ detected by the TGM are shown in the shaded section and enlarged in the breakout panel beneath; see text in section 2.3.2 for details.**

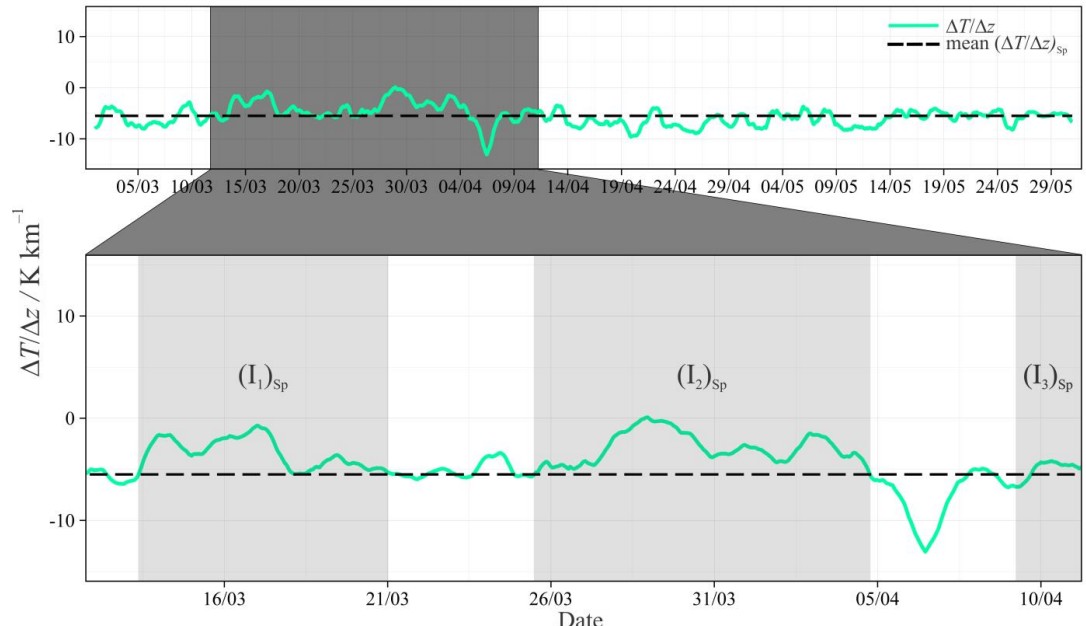

**Figure 12: 24-h running mean of the pseudo-vertical temperature gradient in spring; three 'persistent temperature inversion' events $(I_i)_{Sp}$ detected by the TGM are shown in the shaded area and enlarged in the breakout panel beneath; see text in section 2.3.2 for details.**





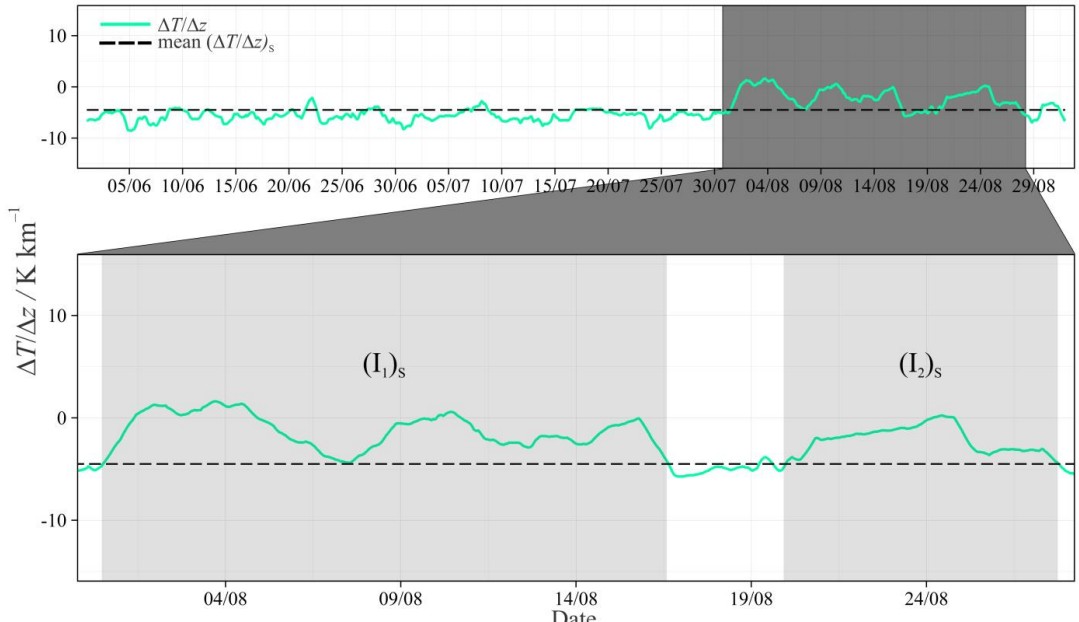

**Figure 13: 24-h running mean of the pseudo-vertical temperature gradient in summer; two 'persistent temperature inversion' events ($I_i$)s detected by the TGM are shown in the shaded section and enlarged in the breakout panel beneath; see text in section 2.3.2 for details.**

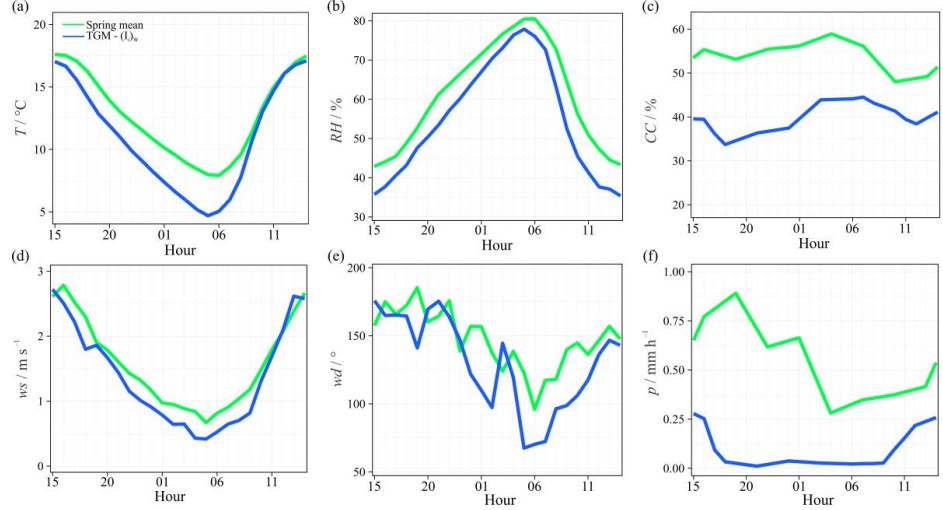

**Figure 14: Diurnal composites, based on hourly means, of: (a) air temperature, (b) relative humidity, (c) cloud cover, (d) wind speed, (e) wind direction, and (f) precipitation, for spring mean, and 'persistent temperature inversion' events detected in spring by the TGM. No 'persistent temperature inversion' events were detected by the RBM this season.**




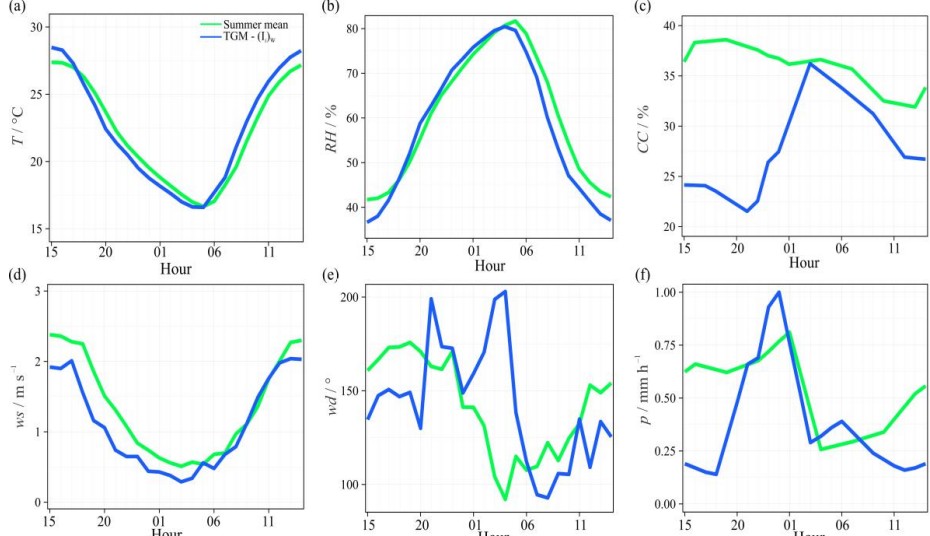

**Figure 15: Diurnal composites, based on hourly means, of: (a) air temperature, (b) relative humidity, (c) cloud cover, (d) wind speed, (e) wind direction, and (f) precipitation, for summer mean, and 'persistent temperature inversion' events detected in summer by the TGM. No 'persistent temperature inversion' events were detected by the RBM this season.**

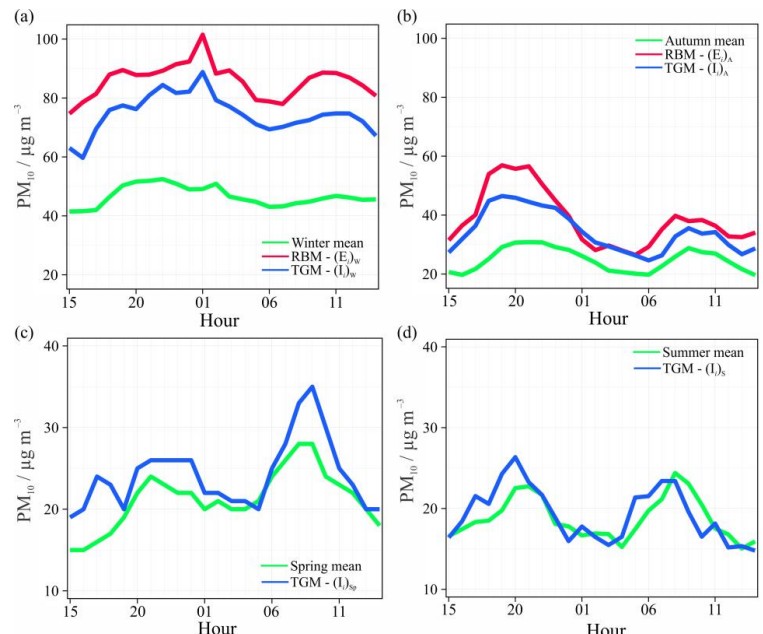

**Figure 16: Diurnal PM$_{10}$ composites, based on hourly means, for: (a) winter, (b) autumn, (c) spring, and (d) summer, for all conditions (seasonal mean) and for 'persistent temperature inversion' events detected by the RBM and TGM.**