# Peer review of "Identifying 'persistent temperature inversion' events in a Subalpine Basin using Radon-222"

_Atmospheric Measurement Techniques, 2018_

## Referee Comment (RC1) · Roberto Salzano (Referee) · 22 Mar 2019

The study is focused on the identification of PTI events in a subalpine basin using two different methods: a radon-based approach and a pseudo-vertical temperature gradient technique. The first part of the paper is aimed to describe the efficacy of the radon-based approach, the second section on the comparison with the second method and finally the evaluation of observations in relation to the air quality.

I have three major comments that affect the manuscript reading:

**1 The radon analysis includes a long timescale contribution. Authors state that seasonal variability [P3-Line 33-36] depends on soil emanation during the year and they remove this contribution dividing the year using a calendar criteria. Recent papers**

[Figure]

[Salzano et al 2016, Salzano et al 2018] modeled 222Rn emanation in agreement with observed variations [Szegvary et al 2007, Szegvary et al 2009, Zhuo et al 2008] and they showed that variations of soil emanation can be significant at a seasonal scale (up two or three times higher in summer compared to winter depending on latitude and site climatology) and probably also at a synoptic and daily scale depending on precipitations and soil freezing (10-20% higher when soils are dryer, for example at mid day). Season identification can be supported by the definition of soil emanating conditions. Authors could add at least precipitations to figure 5 and they could highlight soil-moisture effects that can affect monthly statistics, for example in September.

**2 Figures and text must be re-organized in order to increase readability. Please re-organize the text and the figures in order to group RBM and TGM results without jumping ahead and back from results to discussions.**

**3 The identification of PTI events with RBM is based on the selection of a threshold defined statistically using the standard deviation of the synoptic 222Rn. Figures 2,6,7,8 show a dashed line but it is not clear in the text how it is calculated this value. Is it one for the whole year? Different values for each figure? Can the selection of the period impact on the threshold estimation?**

In details:

P4-Line 25 The radon element (IUPAC) has different isotopes and only 222Rn is the decay product of 226Ra. Please refer to the IUPAC definition of radon as the element in the whole manuscript.

P5 Lines 27-33 What are you measuring, Rn or 222Rn? What is the Alphaguard model?

Figure 2 I suggest authors to show the whole dataset in order to describe longer timescale radon contributions (Figure 5a ???). While the synoptic variability is clearly recognizable from the picture (red line versus black line), the next step is hard to be

[Figure]

understood and replicated. You could, for example, overlay an additional line for the long-term subtraction.

Figure 2b shows a subtraction of 222Rn that comes from the analysis described by figure 3b. The amount of this subtraction in the different seasons are partially described in the text and in the processing data chain (section 2.3.1) but numbers are not reported anywhere.

Figures 2, 6, 7, 8 should be prepared together with the same 222Rn scale (nor Rn). The title in the "a" figure is OK but must defined differently from titles in figures "b". There is subtraction and only in P7-Line 28 and later in figure 5b you specify what you mean with "synoptic".... I suggest also to specify over the dashed line the value of sigma, it is not possible to define numbers from the y-axis. Colours and captions can help readers.

Figure 3a should be prepared for all the seasons as well as 3b.

Figure 5 should be moved before and colours of diurnal and synoptic 222Rn could be used also in figures 2,6,7,8

Figures 11,12,13 should stay with figure 4 in section 3.3

Figures 9,10,14,15 should stay together as well as the specific text in section 3.4

REFERENCES Salzano et al 2016 DOI: 10.1007/s10546-016-0149-6 Salzano et al 2018 DOI: 10.5194/acp-18-6959-2018 Szegvary et al 2007 DOI: 10.1016/j.jenvrad.2008.01.011 Szegvary et al 2009 DOI:10.1016/j.atmosenv.2008.11.025 Zhuo et al 2008 www.atmos-chem-phys.net/7/2789/2007/

---

## Referee Comment (RC2) · Anonymous Referee #1 · 22 Mar 2019

In this paper the authors provide an efficient methodological strategy to solve PBL problems , including the characterization events of extreme stability promoting the buildup of excess air pollution levels. The approach presented is based on the smart use of radon, a historical atmospheric radiotracer, including an extremely accurate evaluation of radon data within a very detailed meteorological framework. The methodology developped shows a great potential in capturing the boundary layer behaviour in districts of complex topography. The method efficiently succeeds in detecting critical stability events whereas other techniques present huge limitations both in the description of the event and in the predictability potential. The comparison with a temperature based methodology proposed by other authors and applied simultaneously with radon appears less precise and reliable than the radon-based approach, while in complex

topography and non uniform landscapes, it is known that boundary layer is poorly described by atmospheric modelling whose limited spatial resolution prevent for suitable description. Improvements in the methdology are expected with the introduction on a routine basis of more sensitive devices for radon measurement (herein defined as "research-oriented": I would say "atmosphere-oriented) in respect to the classical Alphaguard herein used. I do agree that the extensive use of this method would greatly improve the study of boundary layer problems, even when atmospheric modelling will reach sufficient capability. As such radon, thanks to its intrinsic characteristics, will always be useful not only for indepent but objective evaluation of the PBL conditions, but also in other non negligible aspects such as the model calibration.

―――――――――――――――――――

---

## Referee Comment (RC3) · Anonymous Referee #1 · 22 Mar 2019

I had the same objection on the precipitation influence as Dr. Salzano, but the authors' reply was acceptable to me on a first approximation, given the limited precipitation of the period in which the experiment was carried out (use of a seasonal average and inherent smoothing in radon emission) though I agree that assessment in flux variability would be necessary in case of higher precipitation rates. I agree with Dr. Salzano also with the use of a more rigorous terminology: 222Rn instead of radon. I'd like to point out however that the use of Alphaguard contains implicitly the fact that this instrument works on alpha spectrometry enabling the measurement of 222Rn and 220Rn separately though this is possible in the indoor or in soil radon assessment, while the device is not the best for atmospheric observation as commented by the authors. In this experiment therefore Radon is 222Rn , and Alphaguard (which is one of the most worldwide

popular instruments in radon monitoring) does not yield thoron data for the insufficient detection limits in air; in any case if measurable it would be resolved. An example of "atmosphere oriented" radon instruments with similar physical principle is the following: Tositti, L., Pereira, E.B., Sandrini, S., Capra, D., Tubertini, O., Bettoli, M.G. Assessment of summer trends of tropospheric radon isotopes in a coastal Antarctic station (Terra Nova Bay) (2002) International Journal of Environmental Analytical Chemistry, 82 (5), pp. 259-274. In this work not only radon main isotopes are obviously resolved based on the use of a home-made instrument lodging a highly sensitive silicon detector for alpha spectrometry, but both isotopes may be suitably applied for air masses identification even at extremely low concentration levels such as in Antarctica.

---

## Author Comment (AC1) · 14 May 2019

Dafina Kikaj, PhD Candidate
Jožef Stefan International Postgraduate School
Ljubljana, Slovenia

**14 May 2019**

Dr Robyn Schofield, Associate Editor
Atmospheric Measurement Techniques

Dear Dr Schofield,

Please find below our detailed responses to questions from Dr Roberto Salzano regarding our paper: "Identifying 'persistent temperature inversion' events in a Subalpine Basin using Radon-222".

We would like to thank Dr Salzano for his constructive feedback and suggested additional reference material. All comments are addressed individually below.

Best regards,

Dafina Kikaj
(on behalf of all co-authors)

**Response to Roberto Salzano's comments, salzano@iia.cnr.it**

*I have three major comments that affect the manuscript reading:*

*Comment #1: The radon analysis includes a long timescale contribution. Authors state that seasonal variability [P3-Line 33-36] depends on soil emanation during the year and they remove this contribution dividing the year using a calendar criteria. Recent papers [Salzano et al 2016, Salzano et al 2018] modeled $^{222}$Rn emanation in agreement with observed variations [Szegvary et al 2007, Szegvary et al 2009, Zhuo et al 2008] and they showed that variations of soil emanation can be significant at a seasonal scale (up two or three times higher in summer compared to winter depending on latitude and site climatology) and probably also at a synoptic and daily scale depending on precipitations and soil freezing (10–20% higher when soils are dryer, for example at mid-day). Season identification can be supported by the definition of soil emanating conditions. Authors could add at least precipitations to figure 5 and they could highlight soil-moisture effects that can affect monthly statistics, for example in September.*

**Thank you for proposing additional references regarding the $^{222}$Rn exhalation. Indeed, they are relevant and important for our study. The references have been added in the revised manuscript.**
**In order to show possible influences of soil moisture and freezing on the radon exhalation rate (consequently on the atmospheric radon concentration), a third panel has been added to the revised Figure 5, which contains the total daily precipitation and daily mean temperature for the whole year. Also, a discussion of the influences of soil**

**saturation and freezing on the observed radon variability for the four seasons has been added in the revised manuscript.**

*Comment #2: Figures and text must be re-organized in order to increase readability. Please reorganize the text and the figures in order to group RBM and TGM results without jumping ahead and back from results to discussions.*

**As suggested, we have exchanged the order of Figures 2 and 3 to improve the flow of the manuscript. We have also moved the comparative diurnal composite plots of meteorology and PM$_{10}$ to the end of the manuscript. Regarding Figures 2 (Figure 3 in the revised manuscript) and 4 in Methods section, which represent the seasonal examples of each approach of the presented techniques, we would rather leave in this section, because they support the description of the techniques. Moving the Figures 11, 12 and 13 from the Results to Methods section would require total rearrangement of the manuscript and confuse the necessary distinction between describing methods and presenting results.**

*Comment #3: The identification of PTI events with RBM is based on the selection of a threshold defined statistically using the standard deviation of the synoptic 222Rn. Figures 2,6,7,8 show a dashed line but it is not clear in the text how it is calculated this value. Is it one for the whole year? Different values for each figure? Can the selection of the period impact on the threshold estimation?*

**As described on Page 7, Lines 34–36 of the manuscript (and mentioned in each of the figure captions), the threshold value used in Figures 2, 6, 7 and 8 is the standard deviation (+1σ) of the "diurnal radon contribution" in each season (calculated in the standard way over each season) and not the "synoptic radon contribution". It is important for this threshold to change seasonally, since seasonal changes in insolation cause seasonal changes in the diurnal amplitude of the "diurnal radon contribution" (through changes in mixing), which in turn, changes the relative magnitude of the diurnal and seasonal contribution to changes in observed radon.**

**In detail:**

*P4-Line 25 The radon element (IUPAC) has different isotopes and only 222Rn is the decay product of 226Ra. Please refer to the IUPAC definition of radon as the element in the whole manuscript.*

**When the term "radon" was used for the first time in the manuscript (Page 3, Lines 18–19), we put "$^{222}$Rn" in brackets, after which we used the term "radon" or "$^{222}$Rn". Although, when talking about radon, we usually had in our mind the isotope "$^{222}$Rn", we agree, that it is more exact to use the symbol, as we corrected throughout the manuscript.**

*P5 Lines 27-33 What are you measuring, Rn or 222Rn? What is the Alphaguard model?*

**We are using an AlphaGUARD PQ2000 PRO operating in *diffusion* mode (not flow mode). Based on the findings of Kochowska et al., (2009), since we are measuring in a Stevenson's Screen 1.5 m from the surface, it is likely that $^{220}$Rn (thoron) contributes less than 5% to the radon concentrations we report in this study. Therefore, to a good**

**approximation, we can say that we are measuring $^{222}$Rn since the uncertainty on an hourly AlphaGUARD radon measurement even at concentrations as high as 20 Bq m$^{-3}$ is around 15%, and increases with decreasing concentration (Westphal, 2018). In the revised manuscript, the model of the AlphaGUARD is added.**

*Figure 2 I suggest authors to show the whole dataset in order to describe longer timescale radon contributions (Figure 5a ???). While the synoptic variability is clearly recognizable from the picture (red line versus black line), the next step is hard to be understood and replicated. You could, for example, overlay an additional line for the long-term subtraction.*

**Since the Figure 2 (Figure 3 in the revised manuscript) intends only to be an example of the technique application (different for each season) in the Methods section, we would rather leave Figure 5a (which shows the entire year of $^{222}$Rn observations) in the Results section (the first figure there), than to bring it (the data overview) into the Methods section where it is not relevant. As suggested, a line (in blue), which indicates the long-term changes in $^{222}$Rn concentration for subtraction, has been added to Figure 2a (in revised manuscript it is Figure 3a).**

*Figure 2b shows a subtraction of 222Rn that comes from the analysis described by figure 3b. The amount of this subtraction in the different seasons are partially described in the text and in the processing data chain (section 2.3.1) but numbers are not reported anywhere.*

**The values of the subtracted radon contributions are 4.2 ±σ2.1 Bq m$^{-3}$, 3.1±σ0.5 Bq m$^{-3}$, 3.8±σ0.7 Bq m$^{-3}$ and 4.8±σ1.3 Bq m$^{-3}$, for winter, spring, summer and autumn, respectively, and are now reported in the revised manuscript. Lines (in blue), indicating these subtracted contributions, are shown in revised Figures 2, 6, 7 and 8.**

*Figures 2, 6, 7, 8 should be prepared together with the same 222Rn scale (nor Rn). The title in the "a" figure is OK but must defined differently from titles in figures "b". There is subtraction and only in P7-Line 28 and later in figure 5b you specify what you mean with "synoptic"... I suggest also to specify over the dashed line the value of sigma, it is not possible to define numbers from the y-axis. Colours and captions can help readers.*

**The positions of Figures 2 and 4 have already been discussed. Regarding y-axis scales, as suggested, have been unified for observed radon variability, as well as for synoptic radon variability in all mentioned figures. Also, the +1σ value has been printed on each of the threshold lines in the figures.**

*Figure 3a should be prepared for all the seasons as well as 3b.*

**The purpose of Figure 3a is to characterize the diurnal cycle of radon variability, for which extremes of amplitude are represented in the summer and winter plots. The important features of these plots are the afternoon minimum and early morning maximum. Adding and labelling all four seasons would cause the curves to overlap and make the figure less informative. Therefore, we would prefer to leave the Figure 3a in its current form. In the Figure 3b, we added the power spectra analysis for the summer season (representing the other seasonal extreme).**

*Figure 5 should be moved before and colours of diurnal and synoptic 222Rn could be used also in figures 2,6,7,8.*

**As previously explained, according to our opinion, a 1-year summary of the radon observations best fit in the beginning of the Results section (not relevant for the methods). Regarding Figures 2, 6, 7, and 8, there are not shown the diurnal radon contributions in these plots, but only the observed (total radon) and synoptic radon contributions.**
**The colour for the synoptic radon contribution (red) is already the same in Figure 5b as it is in Figures 2, 6, 7 and 8.**

*Figures 11,12,13 should stay with figure 4 in section 3.3*

**According to our concept of the manuscript, as previously discussed, a graphical example is needed in the Methods section to support and clarify the new method. For this reason, we would prefer not to move key figures 11, 12 and 13 from the Results to the Methods section, or explanation of the method into the Results section.**

*Figures 9,10,14,15 should stay together as well as the specific text in section 3.4*

**We completely agree with this suggestion which greatly improved the flow of the paper. Figures 9, 10, 14 and 15, comparing meteorology between the 2 methods, now appear together. In the revised text, sections 3.2 and 3.3 discuss the number and time of identified PTI events in each season. All information about the meteorological comparisons have been moved from sections 3.2 and 3.3 to section 3.4.**

**References**

Kochowska, E., K. Kozak, B. Kozłowska, J. Mazur and J. Dorda. Test measurements of thoron concentration using two ionization chambers AlphaGUARD vs. radon monitor RAD7. *Nukleonika,* 54(3):189−192, 2009.

Westphal, Michael. Radon as a tracer in air quality monitoring. Technical University Dresden, Faculty of Environmental Sciences. PhD Thesis, 2018.

---

## Author Comment (AC2) · 14 May 2019

Dafina Kikaj, PhD Candidate
Jožef Stefan International Postgraduate School
Ljubljana, Slovenia

**14 May 2019**

Dr Robyn Schofield, Associate Editor
Atmospheric Measurement Techniques

Dear Dr Schofield,

Please find below our response to the comment from the anonymous reviewer of our paper: "Identifying 'persistent temperature inversion' events in a Subalpine Basin using Radon-222".

We would like to thank the reviewer for taking the time to look over our manuscript and provide us with comprehensive constructive feedback.

Best regards,

Dafina Kikaj
(on behalf of all co-authors)

**Response to Anonymous reviewer's comments**

*In this paper the authors provide an efficient methodological strategy to solve PBL problems, including the characterization events of extreme stability promoting the buildup of excess air pollution levels. The approach presented is based on the smart use of radon, a historical atmospheric radiotracer, including an extremely accurate evaluation of radon data within a very detailed meteorological framework. The methodology developped shows a great potential in capturing the boundary layer behaviour in districts of complex topography. The method efficiently succeeds in detecting critical stability events whereas other techniques present huge limitations both in the description of the event and in the predictability potential. The comparison with a temperature based methodology proposed by other authors and applied simultaneously with radon appears less precise and reliable than the radon-based approach, while in complex topography and non uniform landscapes, it is known that boundary layer is poorly described by atmospheric modelling whose limited spatial resolution prevent for suitable description. Improvements in the methodology are expected with the introduction on a routine basis of more sensitive devices for radon measurement (herein defined as "research-oriented": I would say "atmosphere-oriented) in respect to the classical Alphaguard herein used. I do agree that the extensive use of this method would greatly improve the study of boundary layer problems, even when atmospheric modelling will reach sufficient capability. As such radon, thanks to its intrinsic characteristics, will always be useful not only for indepent but objective evaluation of the PBL conditions, but also in other non negligible aspects such as the model calibration.*

**Thank you for your positive evaluation of our manuscript.**

---

## Author Comment (AC3) · 14 May 2019

Dafina Kikaj, PhD Candidate
Jožef Stefan International Postgraduate School
Ljubljana, Slovenia

**14 May 2019**

Dr Robyn Schofield, Associate Editor
Atmospheric Measurement Techniques

Dear Dr Schofield,

Please find below our detailed responses to the comments from the anonymous reviewer of our paper: "Identifying 'persistent temperature inversion' events in a Subalpine Basin using Radon-222".

We would like to thank the reviewer for taking the time to look over our manuscript and provide us with comprehensive constructive feedback.

Best regards,

Dafina Kikaj
(on behalf of all co-authors)

**Response to Anonymous reviewer's comments**

*Comment #1: I had the same objection on the precipitation influence as Dr. Salzano, but the authors' reply was acceptable to me on a first approximation, given the limited precipitation of the period in which the experiment was carried out (use of a seasonal average and inherent smoothing in radon emission) though I agree that assessment in flux variability would be necessary in case of higher precipitation rates.*

**Indeed, we agree that in some extreme cases (as we have seen, for example, in Finland), seasonal changes in soil moisture and/or freezing can have such a large influence on radon emission that the radon-based method of diurnal timescale stability assessment can't be applied in some seasons. In most cases, however, as demonstrated by Williams et al. (2016) and Wang et al. (2016), even relatively substantial seasonal changes in radon emission can be effectively taken into account.**

*Comment #2: I agree with Dr Salzano also with the use of a more rigorous terminology: 222Rn instead of radon. I'd like to point out however that the use of Alphaguard contains implicitly the fact that this instrument works on alpha spectrometry enabling the measurement of 222Rn and 220Rn separately though this is possible in the indoor or in soil radon assessment, while the device is not the best for atmospheric observation as commented by the authors. In this experiment therefore Radon is 222Rn, and Alphaguard (which is one of the most worldwide popular instruments in radon monitoring) does not yield thoron data for the insufficient detection limits in air; in any case if measurable it would be resolved.*

As mentioned in our reply above to a similar comment by Dr Salzano, it is certainly our intention in this manuscript that use of the word "radon" is understood to mean $^{222}$Rn. We have clarified this in the revised text. Using an AlphaGUARD to monitor outdoor environmental atmospheric radon concentrations (which is not ideal, but economical), the typical uncertainty for an hourly measurement ranges from around 15% at concentrations of 20 Bq m$^{-3}$, to >50% at concentrations of ~3 Bq m$^{-3}$ (Westphal, 2018). Kochowska et al. (2009) indicate that an AlphaGUARD operating in diffusion mode (as was the case for our experiment) only registers around 5% of ambient $^{220}$Rn (thoron) concentrations when exposed to an equilibrium amount of thoron. Given that our instrument was operating 1.5 m above the surface, within a Stevenson's Screen enclosure, it is likely that less than 5% of its reported radon concentration would be attributable to thoron (as the reviewer points out, compared to the instruments uncertainty, this amount is negligible; as is the ~1% contribution of $^{219}$Rn to outdoor ambient radon concentrations). Therefore, to a good approximation, the reported radon concentrations of our study are exclusively $^{222}$Rn.

*Comment #3: An example of "atmosphere oriented" radon instruments with similar physical principle is the following: Tositti, L., Pereira, E.B., Sandrini, S., Capra, D., Tubertini, O., Bettoli, M.G. Assessment of summer trends of tropospheric radon isotopes in a coastal Antarctic station (Terra Nova Bay) (2002) International Journal of Environmental Analytical Chemistry, 82 (5), pp. 259-274. In this work not only radon main isotopes are obviously resolved based on the use of a home-made instrument lodging a highly sensitive silicon detector for alpha spectrometry, but both isotopes may be suitably applied for air masses identification even at extremely low concentration levels such as in Antarctica.*

We agree that a radon detector of the kind constructed and implemented by Tositti et al. (2002), which is capable of independently resolving contributions to ambient radon by $^{222}$Rn and $^{220}$Rn progeny through alpha spectrometry, would be more suitable than the commercial AlphaGUARD unit for most ambient environmental atmospheric monitoring purposes, if sufficient resources are available to acquire such an instrument. Unfortunately, this was not an option for our study.

*Comment #4: Improvements in the methodology are expected with the introduction on a routine basis of more sensitive devices for radon measurement (herein defined as "research-oriented": I would say "atmosphere-oriented) in respect to the classical Alphaguard herein used. I do agree that the extensive use of this method would greatly improve the study of boundary layer problems, even when atmospheric modelling will reach sufficient capability.*

We agree that as more high-quality, routine atmospheric radon (and radon progeny) observations become available in regions of complex topography, this technique could be further investigated and refined, and a range of other applications investigated.

**References**

Kochowska, E., K. Kozak, B. Kozłowska, J. Mazur and J. Dorda. Test measurements of thoron concentration using two ionization chambers AlphaGUARD vs. radon monitor RAD7. *Nukleonika,* 54(3):189−192, 2009.

Wang F, Chambers SD, Zhang Z, Williams AG, et al. Quantifying stability influences on air pollution in Lanzhou, China, using a radon-based "stability monitor": Seasonality and extreme events. *Atmospheric Environment*, 145, 376−391, 2016.

Westphal, Michael. Radon as a tracer in air quality monitoring. Technical University Dresden, Faculty of Environmental Sciences. PhD Thesis, 2018.

Williams AG, Chambers SD, Conen F, Reimann S, Hill M, Griffiths AD and Crawford J. Radon as a tracer of atmospheric influences on traffic-related air pollution in a small inland city. *Tellus B* 68, 30967, 2016, http://dx.doi.org/10.3402/tellusb.v68.30967.

---

## Author Response (AR2)

Dafina Kikaj, PhD Candidate
Jožef Stefan International Postgraduate School
Ljubljana, Slovenia

**22 July 2019**

Dr Robyn Schofield, Associate Editor
Atmospheric Measurement Techniques

Dear Dr Schofield,

Please find below our detailed responses to the technical corrections requested by Dr Roberto Salzano regarding our paper: "Identifying 'persistent temperature inversion' events in a Subalpine Basin using Radon-222".

We would like to thank Dr Salzano for his constructive feedback. All comments are addressed individually below.

Best regards,

Dafina Kikaj
(on behalf of all co-authors)

**Response to Roberto Salzano's comments, salzano@iia.cnr.it**

*I would thank authors for the effort in revising the manuscript and I would highlight some minor comments that could increase the readability of the manuscript to a not-specialized reader.*

*Figure 3 Please show dates in the x-axis of panels 3a and 3b, the x-axis of panel 3c can confuse the reader since the number 12 (December) could the beginning, the middle or the end of the month...*

**As suggested, to reduce ambiguity we have changed the date format on the x-axis of Figures 3, 4, 6–11.**

*Figure 5, there is a lack of data in December? Please specific in section 2.3 line 27 the date and fix the dates in x-axis of figures 3,4,6-11. Fig. 3-4 winter (1 Dec - 28 Feb), Fig.7-10 spring (1 Mar - 31 May), Fig. 8-11 summer (1 Jun - 31 Aug) and Fig. 6-9 autumn (1 Sep – 30 Nov).*

**Yes, there is a lack of data in December, the measurements have started on 14th of December 2016. We have specified the dates in the section 2.3 and also in the x-axis of Figures 3, 4, 6–11.**

*Figures 3,6-8, I should be able to connect easily the defined variables to the text of section 2.3.1... While from figure 3a I find CRn-observed, CRn-advected and CRn-subtracted, in panel 3b I read CRn-synoptic and CRn-diurnal. These variables are cited in the caption but they cannot be found easily in the text:*

*is CRn-observed cited in page 6 line 36?*

*is CRn-advected cited in page 7 line 16? I'm pretty sure and I ask to indicate the variable there...*

*is CRn-subtracted cited in page 7 line 29? Yes, please use the same symbol (substracted CRn or CRn-subtracted)*

*is CRn-synoptic cited in page 7 line 30? Yes, please use the same symbol (synoptic CRn or CRn-synoptic). I understand also that "synoptic" = "advected" - "subtracted"...I think that the declaration of synoptic could be supported by an equation...Please change the color of synoptic in figure 3 and similar...It can be confused with advected...*

*is CRn-diurnal cited in page 7 line 11?*

**Thank you for pointing out this ambiguity. In the final version of manuscript, we have used the same symbols in the text as are in the figure captions ($C_{Rn-observed}$, $C_{Rn-advected}$, $C_{Rn-subtracted}$, $C_{Rn-diurnal}$). The colour of $C_{Rn-advected}$ is changed from light red to light blue.**

*Figure 3b authors show $+\sigma$ (CRn-diurnal) in the legend and then $\sigma$ just above the dashed line, I think they are the same....Call them in the same way or remove $\sigma$ and indicate only the value...*

**The $+\sigma$ has been printed on each of the threshold lines (dashed lines) the same as it is in the legend $+\sigma$ ($C_{Rn-diurnal}$) of graphs.**

*Pay attention to $\sigma$ and $\mu$ in Figures 3,4,6-11 since $\sigma$ and $\mu$ are described also in page 7 line 31 and page 9 line 17.*

**Done. Thank you.**

*P1 Line 12 ... new radon-based method (RBM), based on single-height 222Rn measurements...*

**Done.**

*P4 line 27 what do you mean with "($\pm25\%$; Jacob et al 1997" ? Radon emanation in the Earth crust?*

**For the clarification we have rewritten the sentence as follows: "(1 atom cm$^{-2}$ s$^{-1}\pm25\%$; Jacob et al 1997…"**

*P5 line 10 Julian and Kamnik-Savinje Alps?*

**Done. Thank you.**